# AN OPERATOR PRECONDITIONING PERSPECTIVE ON TRAINING IN PHYSICS-INFORMED MACHINE LEARNING

**Tim De Ryck**[*]
Seminar for Applied Mathematics,
ETH Zürich, Switzerland

**Florent Bonnet**
Institute of Intelligent Systems and Robotics,
Extrality,
Sorbonne Université, France

**Siddhartha Mishra**
Seminar for Applied Mathematics,
ETH AI Center,
ETH Zürich, Switzerland

**Emmanuel de Bézenac**[*]
Seminar for Applied Mathematics,
ETH Zürich, Switzerland

## ABSTRACT

In this paper, we investigate the behavior of gradient descent algorithms in physics-informed machine learning methods like PINNs, which minimize residuals connected to partial differential equations (PDEs). Our key result is that the difficulty in training these models is closely related to the conditioning of a specific differential operator. This operator, in turn, is associated to the *Hermitian square* of the differential operator of the underlying PDE. If this operator is ill-conditioned, it results in slow or infeasible training. Therefore, preconditioning this operator is crucial. We employ both rigorous mathematical analysis and empirical evaluations to investigate various strategies, explaining how they better condition this critical operator, and consequently improve training.

## 1 INTRODUCTION

Partial Differential Equations (PDEs) are ubiquitous as mathematical models of interesting phenomena in science and engineering (Evans, 2010). Traditionally, numerical methods such as finite difference, finite element etc (Quarteroni & Valli, 1994) are used to simulate PDEs. However, given the prohibitive cost of these methods for a variety of PDE problems such as those with multiple scales, in high dimensions or involving multiple calls to the PDE solver like in UQ, control and inverse problems, machine learning based alternatives are finding increasing traction as efficient PDE simulators, see Karniadakis et al. (2021) and references therein.

Within the plethora of approaches that leverage machine learning techniques to solve PDEs, models which directly incorporate the underlying PDE into the loss function are widely popular. A prominent example of this framework, often referred to as *physics-informed machine learning*, are *physics-informed neural networks* or PINNs (Dissanayake & Phan-Thien, 1994; Lagaris et al., 2000a;b; Raissi et al., 2019), which minimize the PDE residual within the ansatz space of neural networks. Related approaches in which the weak or variational form of the PDE residual is minimized include Deep Ritz (E & Yu, 2018), neural Galerkin (Bruna et al., 2022), variational PINNs (Kharazmi et al., 2019) and weak PINNs (De Ryck et al., 2022). Similarly, PDE residual minimization methods for other ansatz spaces such as Gaussian processes (Raissi & Karniadakis, 2018), Fourier features (Tancik et al., 2020), random features (Ye et al., 2023) etc have also been considered.

Despite the considerable success of PINNs and their afore-mentioned variants in solving numerous types of PDE forward and inverse problems (see Karniadakis et al. (2021); Cuomo et al. (2022) and references therein for extensive reviews), significant problems have been identified with physics-informed machine learning. Arguably, the foremost problem lies with the *training* these frameworks

---

[*]These authors contributed equally to this work.

with (variants of) gradient descent methods (Krishnapriyan et al., 2021; Moseley et al., 2021; Wang et al., 2021a; 2022b). It has been increasingly observed that PINNs and their variants are *slow, even infeasible*, to train even on certain model problems (Krishnapriyan et al., 2021), with the training process either not converging or converging to unacceptably large loss values.

What is the reason behind the issues observed with training physics-informed machine learning algorithms? Empirical studies such as Krishnapriyan et al. (2021) attribute failure modes to the non-convex loss landscape, which is much more complex when compared to the loss landscape of supervised learning. Others like Moseley et al. (2021); Dolean et al. (2023) have implicated the well-known spectral bias (Rahaman et al. (2019)) of neural networks as being a cause for poor training whereas Wang et al. (2021a;b) used infinite-width NTK theory to propose that the subtle balance between the PDE residual and supervised components of the loss function could explain and possibly ameliorate training issues. Nevertheless, it is fair to say that there is a paucity of principled analysis of the training process for gradient descent algorithms in the context of physics-informed machine learning. This provides the context for the current work where we aim to rigorously analyze gradient descent based training in physics-informed machine learning, identify a potential cause of slow training and provide possible strategies to alleviate it. To this end, our main contributions are,

- We derive precise conditions under which gradient descent for a physics-informed loss function can be approximated by a *simplified gradient descent* algorithm, which amounts to the gradient descent update for a *linearized* form of the training dynamics.

- Consequently, we prove that the speed of convergence of the gradient descent is related to the *condition number* of an operator, which in turn is composed of the *Hermitian square* ($\mathcal{D}^*\mathcal{D}$) of the differential operator ($\mathcal{D}$) of the underlying PDE and a *kernel integral operator*, associated to the tangent kernel for the underlying model.

- This analysis automatically suggests that *preconditioning* the resulting operator is necessary to alleviate training issues for physics-informed machine learning.

- By a combination of rigorous analysis and empirical evaluation, we examine how different preconditioning strategies can overcome training bottlenecks and also investigate how existing techniques, proposed in the literature for improving training, can be viewed from this new operator preconditioning perspective.

## 2 ANALYZING TRAINING FOR PHYSICS-INFORMED MACHINE LEARNING IN TERMS OF OPERATOR CONDITIONING.

**Setting.** Our underlying PDE is the following abstract equation,

$$
\begin{aligned}
\mathcal{D}u(x) &= f(x), \quad x \in \Omega, \\
u(x) &= g(x), \quad x \in \partial\Omega.
\end{aligned}
\tag{2.1}
$$

Here, $\Omega \subset \mathbb{R}^d$ is an open bounded subset of either space or space-time, depending on whether the PDE depends on time or not. The PDE (2.1) is specified in terms of the *differential operator* $\mathcal{D}$ and the boundary conditions given by $g$. Specific forms of the differential operator $\mathcal{D}$ are presented later on, whereas for simplicity, we fix Dirichlet-type boundary conditions in (2.1), while other types of boundary conditions can be similarly treated. Finally, we consider the solution $u : \Omega \to \mathbb{R}$ as a scalar for simplicity although all the considerations below also for apply to the case of a vector $u$.

Physics-informed machine learning relies on an *ansatz space* of *parametric functions*, $u(\cdot\,;\theta) : \Omega \mapsto \mathbb{R}$ for all $\theta \in \Sigma \subset \mathbb{R}^n$. This ansatz space could consist of linear (affine) combinations of basis functions $\sum_{k=1}^{n} \theta_k \phi_k$, with possible basis functions as trigonometric functions or finite-element type piecewise polynomial functions or it could consist of nonlinear parametric functions such as neural networks (Goodfellow et al., 2016) or Gaussian processes (Rasmussen, 2003).

The aim is to find parameters $\theta \in \Sigma$ such that the resulting parametric function $u(\cdot\,;\theta) \approx u$, approximates the solution $u$ of the PDE (2.1). In contrast to supervised learning, where the parameters $\theta$ would be chosen to fit (possibly noisy) data $u(x_i)$ with $x_i \in D$, the key ingredient in physics-

informed machine learning is to consider the loss function

$$L(\theta) = \underbrace{\frac{1}{2} \int_\Omega |\mathcal{D}u(x) - f(x)|^2 \, dx}_{R(\theta)} + \underbrace{\frac{\lambda}{2} \int_{\partial\Omega} |u(x) - g(x)|^2 \, d\sigma(x)}_{B(\theta)}, \quad (2.2)$$

with *PDE residual* $R$, supervised loss $B$ at the boundary and a parameter $\lambda > 0$ that relatively weighs the two components of the loss function. In practice, the integrals in the loss function (2.2) need to replaced by suitable quadratures, but as long as the number of quadrature (sampling) points is sufficiently large, the corresponding generalization errors (Mishra & Molinaro, 2020; De Ryck & Mishra, 2021) can be made arbitrarily small.

**Characterization of Gradient Descent for Physics-informed Machine Learning.** Physics-informed machine learning boils down to minimizing the physics-informed loss (2.2), i.e. to find,

$$\theta^\dagger = \underset{\theta \in \Sigma}{\operatorname{argmin}} \, L(\theta). \quad (2.3)$$

Once such an (approximate) minimizer $\theta^\dagger$ is obtained, one appeals to theoretical results such as those in Mishra & Molinaro (2020); De Ryck & Mishra (2021); De Ryck et al. (2021) to show that $u(\cdot \, ; \theta^\dagger)$ approximates the solution $u$ of the PDE (2.1) to high accuracy. Moreover, explicit error estimates in terms of the training error $L(\theta^\dagger)$ can also be obtained (Mishra & Molinaro, 2020; De Ryck & Mishra, 2021). As is customary in machine learning (Goodfellow et al., 2016), the non-convex optimization problem (2.3) is solved with (variants of) a gradient descent algorithm which takes the following generic form,

$$\theta_{k+1} = \theta_k - \eta \nabla_\theta L(\theta_k), \quad (2.4)$$

with descent steps $k > 0$, learning rate $\eta > 0$, loss $L$ (2.2) and the initialization $\theta_0$ chosen randomly.

Our aim here is to analyze whether this gradient descent algorithm (2.4) *converges* as $k \to \infty$ to a minimizer of (2.3). Moreover, we want to investigate the *rate of convergence* to ascertain the computational cost of training. As the loss $L$ (2.4) is non-convex, it is hard to rigorously analyze the training process in complete generality. One needs to make certain assumptions on (2.4) to make the problem tractable. To this end, we fix step $k$ in (2.4) and start with the following Taylor expansion,

$$u(x; \theta_k) = u(x; \theta_0) + \nabla_\theta u(x; \theta_0)^\top (\theta_k - \theta_0) + \tfrac{1}{2}(\theta_k - \theta_0)^\top H_k(x)(\theta_k - \theta_0). \quad (2.5)$$

Here, $H_k(x) := \mathrm{Hess}_\theta(u(x; \tau_k\theta_0 + (1 - \tau_k)\theta_k)$ is the Hessian of $u(\cdot, \theta)$ evaluated at intermediate values, with $0 \le \tau_k \le 1$. Now introducing the notation $\phi_i(x) = \partial_{\theta_i} u(x; \theta_0)$, and assuming that $\mathcal{D}\phi_i \in L^2(\Omega)$, we define the matrix $\mathbb{A} \in \mathbb{R}^{n \times n}$ and the vector $\mathbb{B} \in \mathbb{R}^n$ as,

$$\begin{aligned} \mathbb{A}_{i,j} &= \langle \mathcal{D}\phi_i, \mathcal{D}\phi_j \rangle_{L^2(\Omega)} + \lambda \langle \phi_i, \phi_j \rangle_{L^2(\partial\Omega)}, \\ \mathbb{B}_i &= \langle f - \mathcal{D}u_{\theta_0}, \mathcal{D}\phi_i \rangle_{L^2(\Omega)} + \lambda \langle u - u_{\theta_0}, \phi_i \rangle_{L^2(\partial\Omega)}. \end{aligned} \quad (2.6)$$

Substituting the above formulas in the GD algorithm (2.4), we can rewrite it identically as,

$$\theta_{k+1} = \theta_k - \eta \nabla_\theta L(\theta_k) = (I - \eta\mathbb{A})\theta_k + \eta(\mathbb{A}\theta_0 + \mathbb{B}) + \eta\varepsilon_k, \quad (2.7)$$

where $\varepsilon_k$ is an error term that collects all terms that depend on the Hessians $H_k$ and $\mathcal{D}H_k$. A full definition and further calculations can be found in **SM** A.1. From this characterization of gradient descent (2.4), we clearly see that (2.4) is related to a *simplified* version of gradient descent given by,

$$\widetilde{\theta}_{k+1} = (I - \eta\mathbb{A})\widetilde{\theta}_k + \eta(\mathbb{A}\widetilde{\theta}_0 + \mathbb{B}), \quad \widetilde{\theta}_0 = \theta_0, \quad (2.8)$$

modulo the *error* term $\varepsilon_k$ defined in (2.7).

In the following Lemma (proved in **SM** A.2), we show that this simplified GD dynamics (2.8) approximates the full GD dynamics (2.4) to desired accuracy as long as the error term $\varepsilon_k$ is small.

**Lemma 2.1.** *Let $\delta > 0$ be such that $\max_k \|\varepsilon_k\|_2 \le \delta$. If $\mathbb{A}$ is invertible and $\eta = c/\max_j |\lambda_j(\mathbb{A})|$ for some $0 < c < 1$ then it holds for any $k \in \mathbb{N}$ that,*

$$\|\theta_k - \widetilde{\theta}_k\|_2 \le \delta/\min_j |\lambda_j(\mathbb{A})|. \quad (2.9)$$

The key assumption in Lemma 2.1 is the smallness of the error term $\varepsilon_k$ (2.7) for all $k$. This is trivially satisfied for linear models $u_\theta(x) = \sum_k \theta_k \phi_k$ as $\varepsilon_k = 0$ for all $k$ in this case. From the definition of $\varepsilon_k$ (**SM** (A.1)), we see that a more general sufficient condition for ensuring this smallness is to ensure that the Hessians of $u_\theta$ and $\mathcal{D}u_\theta$ (resp. $H_k$ and $\mathcal{D}H_k$ in (2.5)) are small during training. This amounts to requiring *approximate linearity* of the parametric function $u(\cdot\,;\theta)$ near the initial value $\theta_0$ of the parameter $\theta$. For any differentiable parametrized function $f_\theta$, its linearity is equivalent to the constancy of the associated *tangent kernel* $\Theta[f_\theta](x, y) := \nabla_\theta f_\theta(x)^\top \nabla_\theta f_\theta(y)$ (Liu et al., 2020). Hence, it follows that if the tangent kernel associated to $u_\theta$ and $\mathcal{D}u_\theta$ is (approximately) constant along the optimization path, then the error term $\varepsilon_k$ will be small.

For neural networks this entails that the *neural tangent kernels* (NTK) $\Theta[u_\theta]$ and $\Theta[\mathcal{D}u_\theta]$ stay approximately constant along the optimization path. The following informal lemma, based on Wang et al. (2022b), confirms that this is indeed the case for wide enough neural networks. A rigorous version of the result and its proof can be found in **SM** A.3.

**Lemma 2.2.** *For a neural network $u_\theta$ with one hidden layer of width $m$ and a linear differential operator $\mathcal{D}$ it holds that $\lim_{m\to\infty} \Theta[u_{\theta_k}] = \lim_{m\to\infty} \Theta[u_{\theta_0}]$ and $\lim_{m\to\infty} \Theta[\mathcal{D}u_{\theta_k}] = \lim_{m\to\infty} \Theta[\mathcal{D}u_{\theta_0}]$ for all $k$. Consequently, the error term $\varepsilon_k$ (2.7) is small for wide neural networks, $\lim_{m\to\infty} \max_k \|\varepsilon_k\|_2 = 0$.*

**Convergence of Simplified Gradient Descent Iterations (2.8).** Given the much simpler structure of (2.8), when compared to (2.4), we can study the corresponding gradient descent dynamics explicitly and obtain the following convergence theorem (proved in **SM**A.4),

**Theorem 2.3.** *Let $\mathbb{A}$ in (2.8) be invertible with condition number $\kappa(\mathbb{A})$,*

$$\kappa(\mathbb{A}) = \lambda_{\max}(\mathbb{A})/\lambda_{\min}(\mathbb{A}) = \max_j |\lambda_j(\mathbb{A})| / \min_j |\lambda_j(\mathbb{A})|, \tag{2.10}$$

*and let $0 < c < 1$. Set $\eta = c/\lambda_{\max}(\mathbb{A})$ and $\theta^* = \theta_0 + \mathbb{A}^{-1}\mathbb{B}$. It holds for any $k \in \mathbb{N}$ that,*

$$\|\widetilde{\theta}_k - \theta^*\|_2 \leq \left(1 - c/\kappa(\mathbb{A})\right)^k \|\theta_0 - \theta^*\|_2. \tag{2.11}$$

An immediate consequence of the quantitative convergence rate (2.11) is as follows: to obtain an error of size $\varepsilon$, i.e., $\|\widetilde{\theta}_k - \theta^*\|_2 \leq \varepsilon$, we can readily calculate the number of GD steps $N(\varepsilon)$ as,

$$N(\varepsilon) = \ln\left(\varepsilon/\|\theta_0 - \theta^*\|_2\right) / \ln\left(1 - c/\kappa(\mathbb{A})\right) = O\left(\kappa(\mathbb{A}) \ln \tfrac{1}{\varepsilon}\right). \tag{2.12}$$

Hence, for a fixed value $c$, large values of the condition number $\kappa(\mathbb{A})$ will severely impede convergence of the simplified gradient descent (2.8) by requiring a much larger number of steps.

**Operator Conditioning.** So far, we have established that, under suitable assumptions, the rate of convergence of the gradient descent algorithm for physics-informed machine learning boils down to the *conditioning* of the matrix $\mathbb{A}$ (2.6). However, at first sight, this matrix is not very intuitive and we want to relate it to the differential operator $\mathcal{D}$ from the underlying PDE (2.1). To this end, we first introduce the so-called *Hermitian square* $\mathcal{A}$ given by $\mathcal{A} = \mathcal{D}^*\mathcal{D}$, in the sense of operators, where $\mathcal{D}^*$ is the *adjoint operator* for the differential operator $\mathcal{D}$. Note that this definition implicitly assumes that the adjoint $\mathcal{D}^*$ exists and the Hermitian square operator $\mathcal{A}$ is defined on an appropriate function space. As an example, consider as differential operator the Laplacian, i.e., $\mathcal{D}u = -\Delta u$, defined for instance on $u \in H^1(\Omega)$, then the corresponding Hermitian square is $\mathcal{A}u = \Delta^2 u$, identified as the *bi-Laplacian* that is well defined on $u \in H^2(\Omega)$.

Next for notational simplicity, we set $\lambda = 0$ in (2.2) and omit boundary terms in the following. Let $\mathcal{H}$ be the span of the functions $\phi_k := \partial_{\theta_k} u(\cdot; \theta_0)$. Define the maps $T : \mathbb{R}^n \to \mathcal{H}, v \to \sum_{k=1}^n v_k \phi_k$ and $T^* : L^2(\Omega) \to \mathbb{R}^n; f \to \{\langle \phi_k, f \rangle\}_{k=1,\ldots,n}$. We define the following scalar product on $L^2(\Omega)$,

$$\langle f, g \rangle_{\mathcal{H}} := \langle f, TT^*g \rangle_{L^2(\Omega)} = \langle T^*f, T^*g \rangle_{\mathbb{R}^n}. \tag{2.13}$$

Note that the maps $T, T^*$ provide a correspondence between the continuous space ($L^2$) and discrete space ($\mathcal{H}$) spanned by the functions $\phi_k$. This continuous-discrete correspondence allows us to relate the conditioning of the matrix $\mathbb{A}$ in (2.6) to the conditioning of the Hermitian square operator $\mathcal{A} = \mathcal{D}^*\mathcal{D}$ through the following theorem (proved in **SM** A.5).

**Theorem 2.4.** *It holds for the operator $\mathcal{A} \circ TT^* : L^2(\Omega) \to L^2(\Omega)$ that $\kappa(\mathbb{A}) \geq \kappa(\mathcal{A} \circ TT^*)$. Moreover, if the Gram matrix $\langle \phi, \phi \rangle_{\mathcal{H}}$ is invertible then equality holds, i.e., $\kappa(\mathbb{A}) = \kappa(\mathcal{A} \circ TT^*)$.*

Thus, we show that the conditioning of the matrix $\mathbb{A}$ that determines the speed of convergence of the simplified gradient descent algorithm (2.8) for physics-informed machine learning is intimately tied with the conditioning of the operator $\mathcal{A} \circ TT^*$. This operator, in turn, composes the Hermitian square of the underlying differential operator of the PDE (2.1), with the so-called *Kernel Integral operator $TT^*$*, associated with the (neural) tangent kernel $\Theta[u_\theta]$. Theorem 2.4 implies in particular that if the operator $\mathcal{A} \circ TT^*$ is ill-conditioned, then the matrix $\mathbb{A}$ is ill-conditioned and the gradient descent algorithm (2.8) for physics-informed machine learning will converge very slowly.

**Remark 2.5.** *One can readily generalize Theorem 2.4 to the setting with boundary conditions (i.e., with $\lambda > 0$ in the loss (2.2)). In this case one can prove for the operator $\mathcal{A} = \mathbb{1}_{\mathring{\Omega}} \cdot \mathcal{D}^*\mathcal{D} + \lambda \mathbb{1}_{\partial\Omega} \cdot \mathrm{Id}$, and its corresponding matrix $\mathbb{A}$ (as in (2.6)) that $\kappa(\mathbb{A}) \geq \kappa(\mathcal{A} \circ TT^*)$ in the general case and $\kappa(\mathbb{A}) = \kappa(\mathcal{A} \circ TT^*)$ if the relevant Gram matrix is invertible. The proof is given in* **SM** *A.6.*

**Remark 2.6.** *It is instructive to compare physics-informed machine learning with standard supervised learning through the prism of the analysis presented here. It is straightforward to see that for supervised learning, i.e., when the physics-informed loss in (2.2) is replaced with the supervised loss $\frac{1}{2}\|u - u_\theta\|^2_{L^2(\Omega)}$ by simply setting $\mathcal{D} = \mathrm{Id}$, the corresponding operator in Theorem 2.4 is simply the kernel integral operator $TT^*$, associated with the tangent kernel as $\mathcal{A} = \mathrm{Id}$. Thus, the complexity in training physics-informed machine learning models is entirely due to the spectral properties of the Hermitian square $\mathcal{A}$ of the underlying differential operator $\mathcal{D}$.*

# 3 PRECONDITIONING AND IMPROVING TRAINING IN PHYSICS-INFORMED MACHINE LEARNING.

Having established in the previous section that, under suitable assumptions, the speed of training physics-informed machine learning models depends on the condition number of the operator $\mathcal{A} \circ TT^*$ or, equivalently the matrix $\mathbb{A}$ (2.6), we now investigate whether this operator is ill-conditioned and if so, how can we better condition it by reducing the condition number. The fact that $\mathcal{A} \circ TT^*$ (equiv. $\mathbb{A}$) is very poorly conditioned for most PDEs of practical interest will be demonstrated both theoretically and empirically below. This makes *preconditioning*, i.e., strategies to improve (reduce) the conditioning of the underlying operator (matrix), a key component in improving training for physics-informed machine learning models.

Intuitively, reducing the condition number of the underlying operator $\mathcal{A} \circ TT^*$ can amount to finding new maps $\widetilde{T}, \widetilde{T}^*$ for which the kernel integral operator $\widetilde{T}\widetilde{T}^* \approx \mathcal{A}^{-1}$, i.e., choosing the architecture and initialization of the parametrized model $u_\theta$ such that the associated Kernel Integral operator $\widetilde{T}\widetilde{T}^*$ is an (approximate) Green's function for the Hermitian square $\mathcal{A}$ of the differential operator $\mathcal{D}$. For an operator $\mathcal{A}$ with well-defined eigenvectors $\psi_k$ and eigenvalues $\omega_k$, the ideal case $\widetilde{T}\widetilde{T}^* = \mathcal{A}^{-1}$ is realized when $\widetilde{T}\widetilde{T}^*\phi_k = \frac{1}{\omega_k}\psi_k$.

**Explicit preconditioning by linearly transforming parameters.** The above ideal case can be achieved by transforming $\phi$ (in (2.6) linearly with a (positive definite) matrix $\mathbb{P}$ such that $(\mathbb{P}^\top \phi)_k = \frac{1}{\sqrt{\omega_k}}\psi_k$, which corresponds to the change of variables $\mathcal{P}u_\theta := u_{\mathbb{P}\theta}$. Assuming the invertibility of $\langle \phi, \phi \rangle_\mathcal{H}$, Theorem 2.4 then shows that $\kappa(\mathcal{A} \circ \widetilde{T}\widetilde{T}^*) = \kappa(\widetilde{\mathbb{A}})$ for a new matrix $\widetilde{\mathbb{A}}$, which can be computed as,

$$\widetilde{\mathbb{A}} := \langle \mathcal{D}\nabla_\theta u_{\mathbb{P}\theta_0}, \mathcal{D}\nabla_\theta u_{\mathbb{P}\theta_0} \rangle_{L^2(\Omega)} = \langle \mathcal{D}\mathbb{P}^\top \nabla_\theta u_{\theta_0}, \mathcal{D}\mathbb{P}^\top \nabla_\theta u_{\theta_0} \rangle_{L^2(\Omega)} = \mathbb{P}^\top \mathbb{A}\mathbb{P}. \quad (3.1)$$

This implies a general approach for preconditioning, namely linearly transforming the parameters of the model, i.e. considering $\mathcal{P}u_\theta := u_{\mathbb{P}\theta}$ instead of $u_\theta$, which corresponds to replacing the matrix $\mathbb{A}$ by its preconditioned variant $\widetilde{\mathbb{A}} = \mathbb{P}^\top \mathbb{A}\mathbb{P}$. The new simplified GD update rule is then $\theta_{k+1} = \theta_k - \eta\widetilde{\mathbb{A}}(\theta_k - \theta_0) + \widetilde{\mathbb{B}}$. Hence, finding $\widetilde{T}\widetilde{T}^* \approx \mathcal{A}^{-1}$, which is the aim of preconditioning, reduces to constructing a matrix $\mathbb{P}$ such that $1 \approx \kappa(\widetilde{\mathbb{A}}) \ll \kappa(\mathbb{A})$. We emphasize that $\widetilde{T}\widetilde{T}^*$ need not serve as the exact inverse of $\mathcal{A}$; even an approximate inverse can lead to significant performance improvements, this is the foundational principle of preconditioning.

**Explicit preconditioning by linearly transforming the gradients.** Given that any positive definite matrix can be written as $\mathbb{P}\mathbb{P}^\top$, linearly transforming the parameters is equivalent to precondi-

tioning the gradient of the loss by multiplying with a positive definite matrix, in the sense:

$$\widehat{\theta}_{k+1} = \mathbb{P}\theta_{k+1} = \mathbb{P}\theta_k - \eta\mathbb{P}\mathbb{P}^\top\nabla_\theta L(\mathbb{P}\theta_k) = \widehat{\theta}_k - \eta\mathbb{P}\mathbb{P}^\top\nabla_\theta L(\widehat{\theta}_k), \tag{3.2}$$

which corresponds to performing gradient descent using the transformed parameters $\widehat{\theta}_k := \mathbb{P}\theta_k$. Hence, parameter transformations are all that are needed in this context.

**Analysis of the impact of preconditioning for the Poisson equation.** As an example, we start with linear parametrized models of the form $u_\theta(x) = \sum_k \theta_k\phi_k(x)$, where $\phi_1, \ldots, \phi_n$ are any smooth functions. A corresponding preconditioned model, as explained above, would have the form $\widetilde{u}_\theta(x) = \sum_k (\mathbb{P}\theta)_k\phi_k(x)$, where $\mathbb{P} \in \mathbb{R}^{n\times n}$ is the preconditioner. We motivate the choice of this preconditioner with a simple, yet widely used example.

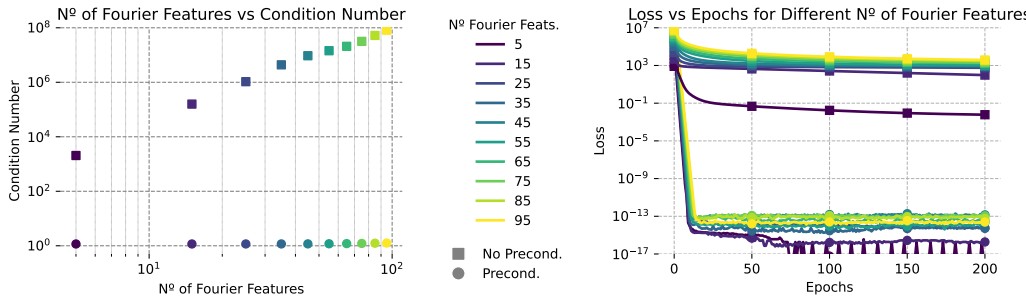

Figure 1: Poisson equation with Fourier features. *Left:* Optimal condition number vs. Number of Fourier features. *Right:* Training for the unpreconditioned and preconditioned Fourier features.

Our differential operator is the one-dimensional Laplacian $\mathcal{D} = \frac{d^2}{dx^2}$, defined on the domain $(-\pi, \pi)$, for simplicity with periodic zero boundary conditions. Consequently, the corresponding PDE (2.1) is the Poisson equation. As the machine learning model, we choose $u_\theta(x) = \sum_{k=-K}^{K} \theta_k\phi_k(x)$, with $\phi_0(x) = \frac{1}{\sqrt{2\pi}}$, $\phi_{-k}(x) = \frac{1}{\sqrt{\pi}}\cos(kx)$ and $\phi_k(x) = \frac{1}{\sqrt{\pi}}\sin(kx)$ for $1 \leq k \leq K$. This model corresponds to the widely used learnable *Fourier Features* in the machine learning literature (Tancik et al., 2020) or *spectral methods* in numerical analysis (Hesthaven et al., 2007). We can readily verify that the resulting matrix $\mathbb{A}$ (2.6) is given by $\mathbb{A} = \mathbb{D} + \lambda vv^\top$, where $\mathbb{D}$ is a diagonal matrix with $\mathbb{D}_{kk} = k^4$ and $v := \phi(\pi)$. Preconditioning solely based on $\mathcal{D}^*\mathcal{D}$ would correspond to finding a matrix $\mathbb{P}$ such that $\mathbb{P}\mathbb{D}\mathbb{P}^\top = \text{Id}$. However, given that $\mathbb{D}_{00} = 0$, this is not possible. We therefore set $\mathbb{P}_{kk} = 1/k^2$ for $k \neq 0$ and $\mathbb{P}_{00} = \gamma \in \mathbb{R}$. The preconditioned matrix is therefore

$$\widetilde{\mathbb{A}}(\lambda, \gamma) = \mathbb{P}\mathbb{D}\mathbb{P}^\top + \lambda\mathbb{P}v(\mathbb{P}v)^\top. \tag{3.3}$$

The conditioning of the unpreconditioned and preconditioned matrices considered above are summarized in the theorem (proved in **SM** B.1) below,

**Theorem 3.1.** *The following statements hold for all $K \in \mathbb{N}$:*

1. *The condition number of the unpreconditioned matrix above satisfies $\kappa(\mathbb{A}(\lambda)) \geq K^4$.*

2. *There exists a constant $C(\lambda, \gamma) > 0$ that is independent of $K$ such that $\kappa(\widetilde{\mathbb{A}}(\lambda, \gamma) \leq C$.*

3. *It holds that $\kappa(\widetilde{\mathbb{A}}(2\pi/\gamma^2, \gamma)) = 1 + O(1/\gamma)$ and hence $\lim_{\gamma \to +\infty} \kappa(\widetilde{\mathbb{A}}(2\pi/\gamma^2, \gamma)) = 1$.*

We observe from Theorem 3.1 that (i) the matrix $\mathbb{A}$, which governs gradient descent dynamics for approximating the Poisson equation with learnable Fourier features, is very poorly conditioned and (ii) we can (optimally) precondition it by *rescaling* the Fourier features based on the eigenvalues of the underlying differential operator (or its Hermitian square).

These conclusions are also observed empirically. In Figure 1 (left), we plot the condition number of the matrix $\mathbb{A}$, minimized over $\lambda$ (see **SM** Figure 8 and SM C for details), as a function of maximum frequency $K$ and verify that this condition number increases as $K^4$, as predicted by the Theorem 3.1. Consequently as shown in Figure 1 (right), where we plot the loss function in terms of increasing training epochs, the underlying Fourier features model is very hard to train with large losses

(particularly for higher values of $K$), showing a very slow decay of the loss function as the number of frequencies is increased. On the other hand, in Figure 1 (left), we also show that the condition number (minimized over $\lambda$) of the preconditioned matrix (3.3) remains constant with increasing frequency and is very close to the optimal value of 1, verifying Theorem 3.1. As a result, we observe from Figure 1 (right) that the loss in the preconditioned case decays exponentially fast as the number of epochs are increased. This decay is independent of the maximum frequency of the model. The results demonstrate that the preconditioned version of the Fourier features model can learn the solution of the Poisson equation efficiently, in contrast to the failure of the unpreconditioned model to do so. Entirely analogous results are obtained with the Helmholtz equation (see **SM** C).

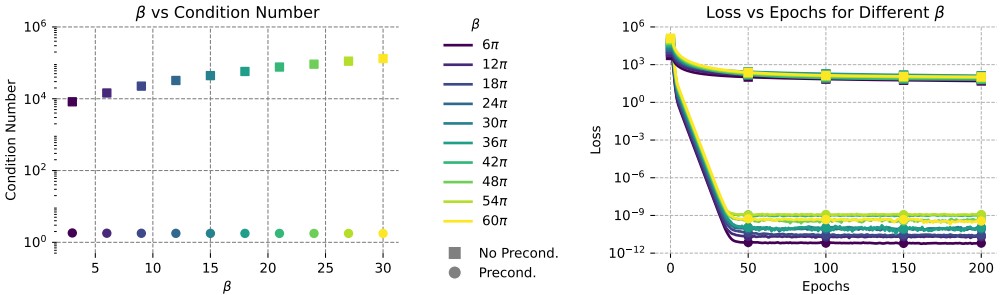

Figure 2: Linear advection equation with Fourier features. *Left:* Optimal condition number vs. $\beta$. *Right:* Training for the unpreconditioned and preconditioned Fourier features.

As a different example, we consider the linear advection equation $u_t + \beta u_x = 0$ on the one-dimensional spatial domain $x \in [0, 2\pi]$ and with $2\pi$-periodic solutions in time with $t \in [0, 1]$. As in Krishnapriyan et al. (2021), our focus in this case is to study how physics-informed machine learning models train when the advection speed $\beta > 0$ is increased. To empirically evaluate this example, we again choose learnable time-dependent Fourier features as the model and precondition the resulting matrix $\mathbb{A}$ (2.6) as described in **SM** B.2.2, see also **SM** C. In Figure 2 (left), we see that the condition number of $\mathbb{A}(\beta) \sim \beta^2$ grows quadratically with advection speed. On the other hand, the condition number of the preconditioned model remains constant. Consequently as shown in Figure 2 (right), the unpreconditioned model trains very slowly (particularly for increasing values of the advection speed $\beta$) with losses remaining high despite being trained for a large number of epochs. In complete contrast, the preconditioned model trains very fast, irrespective of the values of the advection speed $\beta$. Further details including visualizations of the resulting solutions and a comparison with a MLP are presented in **SM** B.2.2 and Figure 13. In particular, we show that the preconditioned Fourier model readily outperforms the MLP. Other additional experiments can be found in **SM** C.

**Viewing available strategies for improving training in physics-informed machine learning models through the lens of operator (pre-)conditioning.** Given the difficulties encountered in training physics-informed machine learning models, several ad-hoc strategies have been proposed in the recent literature to improve training. It turns out that many of these strategies can also be interpreted using the framework of preconditioning that we have proposed. We provide a succinct summary below while postponing the details to the **SM**.

**Choice of $\lambda$.** The parameter $\lambda$ in the loss (2.2) plays a crucial role as it balances the relative contributions of the physics-informed loss $R$ and the supervised loss at the boundary $B$. Given our framework, it is natural to suggest that this parameter should be chosen as $\lambda^* := \min_\lambda \kappa(\mathbb{A}(\lambda))$, in order to obtain the smallest condition number of $\mathbb{A}$ and accelerate convergence. In **SM** B.2, we present $\lambda^*$ for the 1-D Poisson equation with learnable Fourier features to find that $\lambda^*(K) \sim K^2$, with $K$ being the maximum frequency. In turns out that finding suitable values of $\lambda$ has been widely proposed as a strategy, see for instance Wang et al. (2021a; 2022b) which propose algorithms to iteratively learn $\lambda$ during training. It turns out that applying these strategies leads to different scalings of $\lambda$ with respect to increasing $K$ for the Fourier features model (see **SM** B.2 for details), distinguishing our approach for selecting $\lambda$.

**Hard boundary conditions.** From the very advent of PINNs (Lagaris et al., 2000a;b), several authors have advocated modifying machine learning models such that the boundary conditions in PDE (2.1) can be imposed exactly and the boundary loss in (2.2) is zero. Such *hard* imposition of boundary conditions (BCs) has been empirically shown to aid training, e.g. Moseley et al. (2021); Dolean et al. (2023) and references therein. In **SM** B.3, we present an example where the linear advection equation is solved with learnable Fourier features and show that imposing hard BCs reduces the condition number of $\mathbb{A}$, when compared to soft BCs. Thus, hard BCs can improve training by better conditioning the gradient descent dynamics, at least in some cases.

**Second-order optimizers.** There are many empirical studies which demonstrate that first-order optimizers such as (stochastic) gradient descent or ADAM are not suitable for physics-informed machine learning and one needs to use second-order (quasi-)Newton type optimizers such as L-BGFS in order to make training of physics-informed machine learning models feasible. In **SM** B.4, we examine this issue for linear physics-informed models and show that as the Hessian of the loss is identical to the matrix $\mathbb{A}$ (2.6) in this case, (quasi-)Newton methods automatically compute an (approximate) inverse of the Hessian and hence, precondition the matrix $\mathbb{A}$, relating the use of (quasi-)Newton type optimizers to preconditioning operators in this context.

**Domain decomposition.** Domain decomposition (DD) is a widely used technique in numerical analysis to precondition linear systems that arise out of classical methods such as finite elements (Dolean et al., 2015). Recently, there have been attempts to use DD-inspired methods within physics-informed machine learning, see Moseley et al. (2021); Dolean et al. (2023) and references therein, although no explicit link with preconditioning the models was established. In **SM** B.2.2, we re-examine the case of linear advection equation with learnable Fourier features to demonstrate that increasing the number of Fourier features in time by decomposing the time domain simply amounts to changing the effective advection speed $\beta$ and reducing the condition number, leading to a better-conditioned model. Moreover, in this case, this algorithm also correlates the **causal learning** based training of PINNs (Wang et al., 2022a), which also can be viewed as improving the condition number.

## 4 DISCUSSION.

**Summary.** Physics-informed machine learning models are notoriously hard to train with gradient descent methods. In this paper, we aim for a rigorous explanation of the underlying causes as well as examining possible strategies to mitigate them. To this end, under suitable assumptions that coincide with *approximate linearity* of models, we prove that gradient descent with physics-informed losses is approximated well by a novel simplified gradient descent procedure, whose rate of convergence can be completely characterized in terms of the conditioning of an operator, composing the Hermitian square of the underlying differential operator with the Kernel integral operator associated with the underlying tangent kernel. Thus, the ill-conditioning of this Hermitian square operator can explain issues with training of physics-informed learning models. Consequently, *preconditioning* this operator (equivalently the associated matrix) could improve training. By a combination of rigorous analysis and empirical evaluation, we examine strategies with a view of how one can precondition the associated operators. In particular, we find that rescaling the model parameters, as dictated by the spectral properties of the underlying differential operator, was effective in significantly improving training of physics-informed models for the Poisson, Helmholtz and linear advection equations.

**Related Work.** While many studies explore the mathematical aspects of PINNs, the majority focus on approximation techniques or generalization properties (De Ryck & Mishra, 2021; Doumèche et al., 2023). Few works have targeted training error and training dynamics, even though it stands as a significant source of overall error (Krishnapriyan et al., 2021). Some exceptions include Jiang et al. (2023), who examine global convergence for linear elliptic PDEs in the NTK regime. However equations are derived in continuous time, thereby sidestepping ill-conditioning (which is intrinsically linked to discrete time) and thus potential training issues. Wang et al. (2021a) identified that PINNs might converge slowly due to a stiff gradient flow ODE. Our work allows to interpret their proposed novel architecture, which reduces the maximum eigenvalue of the Hessian, as a way to precondition $TT^*$, as the Hessian of the loss equals $\mathbb{A}$ (**SM** B.4), thereby improving the convergence rate (Theorems 2.3 and 2.4). Wang et al. (2022b) derive a continuous-time evolution equation exclu-

sively for the residual during training, leaving out a direct exposition of the Hermitian square term, and contrasting our discrete evolution equation in parameter space, as opposed to function space. Wang et al. (2021a; 2022b) also propose algorithms to adjust the $\lambda$ multiplier between boundary and residual loss terms, which we assess within the context of operator preconditioning in **SM** B.2. Works aiming to improve convergence of PINNs based on domain decomposition strategies include Jagtap & Karniadakis (2020); Jagtap et al. (2020); Wang et al. (2022a); Kopaničáková et al. (2023), some of which can be reinterpreted as methods to precondition $\mathbb{A}$ by changing $\mathcal{A}$ or $TT^*$.

**Limitations and Future Work.** In this work, our examples for elucidating the challenges in training physics-informed machine learning models focussed on linear PDEs. Nevertheless, the analysis already revealed the key role played by equation-dependent preconditioning. Extending our results to nonlinear PDEs is a direction for future work. Moreover, while highlighting the necessity of preconditioning, the current work does not claim to provide a universal preconditioning strategy, particularly for nonlinear models such as neural networks.

We strongly believe that the complications arising from ill-conditioning merit further scrutiny from the scientific computing community, such as those specializing in domain and operator preconditioning. There is much work in this domain (Mardal & Winther, 2011; Hiptmair, 2006) and references therein, providing a fertile ground for innovative approaches, including the potential application of non-linear preconditioning techniques commonly used in these fields. However, extending our work to these settings exceeds the scope of this paper and remains a direction for future inquiry.

Another aspect worth discussing pertains to our linearized training dynamics (NTK regime), in which feature learning is absent (Chizat et al., 2019). For low-dimensional problems typical in many scientific settings (1-3D), the lack of feature learning may not be a significant handicap, as one can discretize the underlying domains. Extensive evidence in this paper has shown that the linear bases often outperform nonlinear models. However, neural networks might still outperform linear models high-dimensional problems (Mishra & Molinaro, 2021), highlighting the significance of deviations from the lazy training regime.

Finally, we would like to point that our analysis can be readily extended to cover physics-informed operator learning models such as those considered in Li et al. (2023); Goswami et al. (2022) by adopting the theoretical framework of representative neural operators (Bartolucci et al., 2023; Raonić et al., 2023).

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

## Supplementary material for:
An operator preconditioning perspective on training in physics-informed machine learning

# A    DETAILS FOR SECTION 2

## A.1    DERIVATION OF EQUATION (2.7)

We provide detailed calculations for how to obtain (2.7). First of all, plugging in (2.5) into (2.2) and calculating yields,

$$\nabla_\theta R(\theta_k) = \int_\Omega (\mathcal{D}u(x;\theta_0) - f(x))\mathcal{D}\nabla_\theta u(x;\theta_0)dx + \int_\Omega \mathcal{D}\nabla_\theta u(x;\theta_0)\mathcal{D}\nabla_\theta u(x;\theta_0)^\top(\theta_k - \theta_0)dx$$
$$+ \frac{1}{2}\int_\Omega (\mathcal{D}u(x;\theta_k) - f(x))\mathcal{D}H_k(\theta_k - \theta_0)dx + \frac{1}{2}\int_\Omega (\theta_k - \theta_0)^\top \mathcal{D}H_k(\theta_k - \theta_0)\mathcal{D}\nabla_\theta u(x;\theta_0).$$

$$\nabla_\theta B(\theta_k) = \int_{\partial\Omega} (u(x;\theta_0) - u(x))\nabla_\theta u(x;\theta_0)dx + \int_{\partial\Omega} \nabla_\theta u(x;\theta_0)\nabla_\theta u(x;\theta_0)^\top(\theta_k - \theta_0)dx$$
$$+ \frac{1}{2}\int_{\partial\Omega} (u(x;\theta_k) - u(x))H_k(\theta_k - \theta_0)dx + \frac{1}{2}\int_{\partial\Omega} (\theta_k - \theta_0)^\top H_k(\theta_k - \theta_0)\nabla_\theta u(x;\theta_0).$$

Now introducing the notation $\phi_i(x) = \partial_{\theta_i} u(x;\theta_0)$, we define the vector $\varepsilon_k \in \mathbb{R}^n$ as,

$$\varepsilon_k = -\frac{1}{2}\Big[\langle \mathcal{D}u_{\theta_k} - f, \mathcal{D}H_k(\theta_k - \theta_0)\rangle_{L^2(\Omega)} + \lambda\langle u_{\theta_k} - u, H_k(\theta_k - \theta_0)\rangle_{L^2(\partial\Omega)}$$

$$+ \langle (\theta_k - \theta_0)^\top \mathcal{D}H_k(\theta_k - \theta_0), \mathcal{D}\nabla_\theta u_{\theta_0}\rangle_{L^2(\Omega)} + \lambda\langle (\theta_k - \theta_0)^\top H_k(\theta_k - \theta_0), \nabla_\theta u_{\theta_0}\rangle_{L^2(\partial\Omega)}\Big].$$
$$\text{(A.1)}$$

Combining this definition with those of $\mathbb{A}$ and $\mathbb{B}$ gives rise to equation 2.7.

## A.2    PROOF OF LEMMA 2.1

*Proof.* If $\max_k \|\varepsilon_k\|_2 \le \delta$ then the error one makes compared to the simplified GD update rule is bounded by

$$\|\theta_l - \widetilde{\theta}_l\|_2 = \left\|\sum_{k=0}^l (I - \eta\mathbb{A})^k \eta\varepsilon_k\right\|_2 \le \eta\sum_{k=0}^l \|I - h\mathbb{A}\|_2^k \|\varepsilon_k\|_2 \tag{A.2}$$

$$\le \eta\sum_{k=0}^l \max_j |1 - \lambda_j(\mathbb{A})\eta|_2^k \delta \le \eta\sum_{k=0}^l (1 - c/\kappa(\mathbb{A}))^k \delta \tag{A.3}$$

$$\le \frac{\kappa(\mathbb{A})\eta\delta}{c} = \frac{\delta}{\min_j |\lambda_j(\mathbb{A})|}. \tag{A.4}$$

$$\square$$

## A.3    PROOF OF LEMMA 2.2

We first provide a rigorous version of the statement of Lemma 2.2 which includes all technical details. To do so, we follow the setting of (Wang et al., 2022b, Section 4). The result is a generalization of (Wang et al., 2022b, Theorem 4.4) that is suggested in Wang et al. (2022b) itself.

**Lemma A.1.** *Let $\mathcal{D}$ be a linear differential operator, $\sigma$ a smooth function and $m, K \in \mathbb{N}$. Let $W^0 \in \mathbb{R}^{1\times m}$, $W^1 \in \mathbb{R}^{m\times 1}$ the weights and $b^0 \in \mathbb{R}^m$ and $b^1 \in \mathbb{R}$ the biases that constitute the parameters $\theta = (W^0, W^1, b^0, b^1)$ of the neural network $u_\theta : \mathbb{R} \to \mathbb{R}$ defined by,*

$$u_\theta(x) = u(x;\theta) = \frac{1}{\sqrt{m}}W^1\sigma(W^0x + b^0) + b^1. \tag{A.5}$$

*Let $(\theta_k)_k$ be the series of parameters one obtains from performing gradient descent on the loss $L(\theta)$, where all entries of the parameters are initialized as iid $\mathcal{N}(0,1)$ random variables. Furthermore, assume that,*

1. *The parameters stay uniformly bounded during training, i.e. there is a constant $C > 0$ independent of $K$ such that $\max_{1 \leq k \leq K} \|\theta_k\|_\infty \leq C$.*

2. *There exists a constant $C > 0$ such that $\sum_{k=1}^{K} L(\theta_k) \leq C$.*

3. *If $\mathcal{D}$ is an $n$-th order differential operator, then there exists a constant $C > 0$ such that $|\sigma^{(l)}(x)| \leq C$ for all $x$ and $0 \leq l \leq 2 + n$.*

*Under these conditions it holds for all $k$ that,*

$$\lim_{m \to \infty} \Theta[u_{\theta_k}] = \lim_{m \to \infty} \Theta[u_{\theta_0}] \quad and \quad \lim_{m \to \infty} \Theta[\mathcal{D}u_{\theta_k}] = \lim_{m \to \infty} \Theta[\mathcal{D}u_{\theta_0}]. \tag{A.6}$$

*Hence, the error term $\varepsilon_k$ (A.1) is small for wide neural networks, $\lim_{m \to \infty} \max_k \|\varepsilon_k\|_2 = 0$.*

*Proof.* In (Wang et al., 2022b, Theorem 4.4) they proved that $\lim_{m \to \infty} \Theta[u_{\theta_k}] = \lim_{m \to \infty} \Theta[u_{\theta_0}]$ and $\lim_{m \to \infty} \Theta[\mathcal{D}u_{\theta_k}] = \lim_{m \to \infty} \Theta[\mathcal{D}u_{\theta_0}]$ in the case that $\mathcal{D} = \partial_{xx}^2$ and state that it can be generalized to more general linear differential operators. We provide the steps to how this more general result can be proven, by summarizing all necessary changes in the proof of (Wang et al., 2022b, Theorem 4.4).

*Changes to(Wang et al., 2022b, Lemma D.1).* Instead of $u_{xx}(x; \theta)$ one needs to provide a formula for $\mathcal{D}u(x; \theta)$. This can be done by noting that,

$$\partial_x^n u(x; \theta) = \frac{1}{\sqrt{m}} \sum_k W_k^1 \sigma^{(n)}(W_k^0 x + b_k^0)(W_k^0)^n, \tag{A.7}$$

where we slightly abuse notation by letting $W_k^i$ and $b_k^i$ be the $k$-th components of the respective vectors, i.e. they do *not* correspond to the $k$-th gradient descent update $\theta_k$. The bounds on $\partial \partial_x^n u(x; \theta) / \partial \theta^l$ then follow in a similar way from the boundedness of both $\theta$ and $\sigma$ and its derivatives.

*Changes to(Wang et al., 2022b, Lemma D.2).* Here we replace the gradient flow by the gradient descent formula $\theta_{k+1} = \theta_k - \eta \nabla_\theta L(\theta_k)$. The rest of the calculations are in the same spirit, with the only difference that we consider a problem that is discrete in (training) time (as we are using gradient descent).

*Changes to(Wang et al., 2022b, Lemma D.4).* The calculations are completely analogous, only that one needs to replace the formulas for $\nabla_\theta u_{xx}(x; \theta)$ by formulas for $\nabla_\theta \mathcal{D}u(x; \theta)$. Again in our simplified case that $\mathcal{D} = \partial_x^n$ we find that,

$$\frac{\partial \mathcal{D}u(x; \theta)}{\partial W_k^0} = \frac{(W_k^0)^{n-1} W_k^1}{\sqrt{m}} \left[ n\sigma^{(n)}(W_k^0 x + b_k^0) + (W_k^0)^2 \sigma^{(n+1)}(W_k^0 x + b_k^0) \right] \tag{A.8}$$

$$\frac{\partial \mathcal{D}u(x; \theta)}{\partial W_k^1} = \frac{1}{\sqrt{m}} \sigma^{(n)}(W_k^0 x + b_k^0)(W_k^0)^n \tag{A.9}$$

$$\frac{\partial \mathcal{D}u(x; \theta)}{\partial b_k^0} = \frac{1}{\sqrt{m}} W_k^1 \sigma^{(n+1)}(W_k^0 x + b_k^0)(W_k^0)^n \tag{A.10}$$

$$\frac{\partial \mathcal{D}u(x; \theta)}{\partial b_k^1} = 0. \tag{A.11}$$

The rest of the proof is a matter of calculations that are analogous to those in Wang et al. (2022b). One can conclude that in the limit $m \to \infty$ the tangent kernel of both $u_\theta$ and $\mathcal{D}u_\theta$ is constant during training. As (approximate) NTK constancy is equivalent to (approximate) linearity Liu et al. (2020), we find that the Hessians $H_k$ and $\mathcal{D}H_k$, and consequently the error term $\varepsilon_k$ must be (approximately) zero. More concretely, the error term $\varepsilon_k$ (A.1) is small for wide neural networks, $\lim_{m \to \infty} \max_k \|\varepsilon_k\|_2 = 0$. $\qquad\square$

## A.4 PROOF OF THEOREM 2.3

*Proof.* We calculate that,

$$\widetilde{\theta}_k = (I - \eta\mathbb{A})\widetilde{\theta}_{k-1} + \eta(\mathbb{A}\widetilde{\theta}_0 + \mathbb{B}) \tag{A.12}$$

$$= (I - \mathbb{A}\eta)^k\widetilde{\theta}_0 + \sum_{\ell=0}^{k-1}(I - \mathbb{A}\eta)^\ell\eta(\mathbb{A}\widetilde{\theta}_0 + \mathbb{B}) \tag{A.13}$$

$$= (I - \mathbb{A}\eta)^k\widetilde{\theta}_0 + (\mathbb{A}\eta)^{-1}(I - (I - \mathbb{A}\eta)^k)\eta(\mathbb{A}\widetilde{\theta}_0 + \mathbb{B}) \tag{A.14}$$

$$= (I - \mathbb{A}\eta)^k\widetilde{\theta}_0 + (I - (I - \mathbb{A}\eta)^k)\theta^* \tag{A.15}$$

and hence

$$\widetilde{\theta}_k - \theta^* = (I - \mathbb{A}\eta)^k(\widetilde{\theta}_0 - \theta^*). \tag{A.16}$$

We can then take the norm, use that $\eta = c/\lambda_{\max}(\mathbb{A})$ and calculate,

$$\|\widetilde{\theta}_k - \theta^*\|_2 \leq \|I - \eta\mathbb{A}\|_2^k\|\widetilde{\theta}_0 - \theta^*\|_2 \leq \max_j |1 - \lambda_j(\mathbb{A})\eta|^k\|\widetilde{\theta}_0 - \theta^*\|_2 \tag{A.17}$$

$$\leq \left(1 - \frac{c\lambda_{\min}(\mathbb{A})}{\lambda_{\max}(\mathbb{A})}\right)^k\|\widetilde{\theta}_0 - \theta^*\|_2 = \left(1 - \frac{c}{\kappa(\mathbb{A})}\right)^k\|\widetilde{\theta}_0 - \theta^*\|_2. \tag{A.18}$$

$\square$

## A.5 PROOF OF THEOREM 2.4

*Proof.* Using that $\mathbb{A} = T^*\mathcal{A}T$ we find,

$$\lambda_{\max}(\mathbb{A}) = \max_{\|v\|=1} v^\top\mathbb{A}v \geq \sup_{f\in L^2:\,\|T^*f\|=1}(T^*f)^\top\mathbb{A}(T^*f) \tag{A.19}$$

$$= \sup_{f\in L^2:\,\|T^*f\|=1}(T^*f)^\top(T^*\mathcal{A}TT^*f) = \sup_{f\in L^2:\,\|T^*f\|=1}\langle f, TT^*\mathcal{A}\circ TT^*f\rangle_{L^2} \tag{A.20}$$

$$= \sup_{f\in L^2:\,\|T^*f\|=1}\langle f, \mathcal{A}\circ TT^*f\rangle_{\mathcal{H}} = \lambda_{\max}(\mathcal{A}\circ TT^*) \tag{A.21}$$

The last equality holds by the minmax theorem since $\mathcal{A}\circ TT^*$ is a self-adjoint operator. Indeed, this operator is self-adjoint since $\mathbb{A}$ is symmetric and given that for $f, g \in \mathcal{H}$ we find that,

$$\langle f, \mathcal{A}\circ TT^*g\rangle_{\mathcal{H}} = (T^*f)^\top\mathbb{A}(T^*g). \tag{A.22}$$

One can do a similar calculation for $\lambda_{\min}$ (with the reverse inequality), which concludes the proof.

Now let $\mathbb{M} = \langle\phi, \phi\rangle_{\mathcal{H}}$ be the Gram matrix, i.e. $\mathbb{M}_{ij} = \langle\phi_i, \phi_j\rangle_{\mathcal{H}}$, and we assume it is invertible. For any parameter vector $v$ we can define $f_v = T\mathbb{M}^{-1}v \in \mathcal{H} \subset L^2(\Omega)$. One can verify that $T^*f_v = v$. As a result we have the chain of inequalities,

$$\sup_{f\in L^2:\,\|T^*f\|=1}(T^*f)^\top\mathbb{A}(T^*f) \leq \max_{\|v\|=1} v^\top\mathbb{A}v \tag{A.23}$$

$$= \max_{\|v\|=1}(T^*f_v)^\top\mathbb{A}(T^*f_v) \tag{A.24}$$

$$\leq \sup_{f\in L^2:\,\|T^*f\|=1}(T^*f)^\top\mathbb{A}(T^*f), \tag{A.25}$$

and hence $\sup_{f\in L^2:\,\|T^*f\|=1}(T^*f)^\top\mathbb{A}(T^*f) = \max_{\|v\|=1} v^\top\mathbb{A}v$. Doing an analogous calculation to the one above, we find that $\kappa(\mathcal{A}\circ TT^*) = \kappa(\mathbb{A})$. $\square$

## A.6 PROOF OF REMARK 2.5

Let $\mathcal{H}$ be the span of the $\phi_k := \partial_{\theta_k}u(\cdot; \theta_0)$. Define the maps $T : \mathbb{R}^n \to \mathcal{H}, v \to \sum_{k=1}^n v_k\phi_k$ and $T^* : L^2(\Omega) \to \mathbb{R}^n; v \to \{\langle\phi_k, f\rangle_{L^2(\Omega)} + \lambda\langle\phi_k, f\rangle_{L^2(\partial\Omega)}\}_{k=1,\ldots,n}$. We define the following scalar product on $L^2(\Omega)$:

$$\langle f, g\rangle_{\mathcal{H}} := \langle f, TT^*g\rangle_{L^2(\Omega)} + \lambda\langle f, TT^*g\rangle_{L^2(\partial\Omega)} = \langle T^*f, T^*g\rangle_{\mathbb{R}^n}. \tag{A.26}$$

We define the operator $\mathcal{A}$ as,

$$\mathcal{A} = \mathbb{1}_{\mathring{\Omega}} \cdot \mathcal{D}^*\mathcal{D} + \lambda \mathbb{1}_{\partial\Omega} \cdot \mathrm{Id}, \tag{A.27}$$

and the matrix $\mathbb{A}$ as,

$$\mathbb{A}_{kl} = \langle \phi_k, \mathcal{D}^*\mathcal{D}\phi_l \rangle_{L^2(\Omega)} + \lambda \langle \phi_k, \phi_l \rangle_{L^2(\partial\Omega)}. \tag{A.28}$$

Let $\mathbb{M} = \langle \phi, \phi \rangle_{\mathcal{H}}$ be the Gram matrix, i.e. $\mathbb{M}_{ij} = \langle \phi_i, \phi_j \rangle_{\mathcal{H}}$. If it is invertible then for any parameter vector $v$ we can define $f_v = T\mathbb{M}^{-1}v \in \mathcal{H} \subset L^2(\Omega)$. One can also verify that $T^*f_v = v$.

We give the proof in the case where $\mathbb{M}$ is invertible. The general case is similar to the proof of Theorem 2.4.

*Proof.* We calculate that for any vector $w \in \mathbb{R}^n$ it holds that,

$$(T^*\mathcal{A}Tw)_k = \langle \phi_k, \mathcal{A}Tw \rangle_{L^2(\Omega)} + \lambda \langle \phi_k, \mathcal{A}Tw \rangle_{L^2(\partial\Omega)} \tag{A.29}$$

$$= \langle \phi_k, \mathcal{D}^*\mathcal{D}Tw \rangle_{L^2(\Omega)} + \lambda \langle \phi_k, Tw \rangle_{L^2(\partial\Omega)} \tag{A.30}$$

$$= \sum_l \left[ \langle \phi_k, \mathcal{D}^*\mathcal{D}\phi_l \rangle_{L^2(\Omega)} + \lambda \langle \phi_k, \phi_l \rangle_{L^2(\partial\Omega)} \right] w_l \tag{A.31}$$

$$= (\mathbb{A}w)_k, \tag{A.32}$$

where we used that the measure of $\partial\Omega$ is zero with respect to the Lebesgue measure on $\Omega$ and that $\mathbb{1}_{\mathring{\Omega}}$ is zero on $\partial\Omega$. Using this and the definition of the scalar product on $\mathcal{H}$ we find that,

$$\lambda_{\max}(\mathbb{A}) = \max_{\|v\|=1} v^\top \mathbb{A}v \tag{A.33}$$

$$= \sup_{\|v\|=1} (T^*f_v)^\top \mathbb{A}(T^*f_v) \tag{A.34}$$

$$= \sup_{\|v\|=1} (T^*f_v)^\top (T^*\mathcal{A}TT^*f_v) \tag{A.35}$$

$$= \sup_{\|v\|=1} \langle f_v, \mathcal{A} \circ TT^*f_v \rangle_{\mathcal{H}} \tag{A.36}$$

$$= \lambda_{\max}(\mathcal{A} \circ TT^*). \tag{A.37}$$

This concludes the proof, as a similar calculation can be done for $\lambda_{\min}(\mathbb{A})$. $\square$

# B   DETAILS FOR SECTION 3

## B.1   PROOF OF THEOREM 3.1

We first the state the following auxiliary result.

**Lemma B.1** (Section 5, Golub (1973)). *Let $\lambda \geq 0$, let $u \in \mathbb{R}^n$, let $D \in \mathbb{R}^{n \times n}$ be a diagonal matrix with eigenvalues $d_i$ such that $d_i \leq d_{i+1}$ and let $C = D + \lambda uu^\top$. The eigenvalues $\omega_1, \ldots, \omega_n$ of $C$ (in ascending order) are the roots of*

$$p(\omega) = \prod_{i=1}^n (d_i - \omega) + \lambda \sum_{i=1}^n u_i^2 \prod_{j=1, j \neq i}^n (d_j - \omega) = 0, \tag{B.1}$$

*and it holds that $d_i \leq \omega_i \leq d_{i+1}$ and $d_n \leq \omega_n \leq d_n + \lambda u^\top u$.*

We now prove Theorem 3.1. In Figure 3 we also plot the condition number of $\widetilde{\mathbb{A}}(\lambda, \gamma)$ for various values of $\lambda, \gamma$ and $K$.

*Proof.* We first prove statement (1). Denote by $\omega_i$ resp. $d_i$ the i-th eigenvalue (in ascending order) of $\mathbb{A}$ resp. $\mathbb{D}$. It follows from Lemma B.1 that $\omega_1 \leq d_2 = 1$ and $\omega_{2K+1} \geq d_{2K+1} = K^4$. Hence we find that for any $\lambda \geq 0$ it holds that,

$$\kappa(\mathbb{A}(\lambda)) = \frac{\omega_{2K+1}}{\omega_1} \geq \frac{d_{2K+1}}{d_2} = K^4. \tag{B.2}$$

We continue with statement (2). It is easy to verify that $\mathbb{P}\mathbb{D}\mathbb{P}$ has eigenvalues 1 (multiplicity $2K$) and 0 (multiplicity 1) and that the vector $v = \phi(\pi)$ has entries $v_{-k} = (-1)^k/\sqrt{\pi}$, $v_0 = 1/\sqrt{2\pi}$, $v_k = 0$ for $1 \le k \le K$. Define $\eta := \sum_{k=1}^{K} 1/k^4$ and note that $1 \le \eta \le \pi^4/90 \approx 1.08$.

Now, by Lemma B.1 the eigenvalues of $\widetilde{\mathbb{A}}$ are given by the roots of,

$$p_{2K+1}(\omega) = -\omega(1-\omega)^{2K} + \frac{\lambda}{\pi}\left(\frac{\gamma^2}{2}(1-\omega)^{2K} - \eta\omega(1-\omega)^{2K-1}\right) \tag{B.3}$$

$$= \frac{1}{2\pi}(1-\omega)^{2K-1}\left(-2\pi\omega(1-\omega) + \lambda\gamma^2(1-\omega) - 2\lambda\omega\eta\right) \tag{B.4}$$

$$= \frac{1}{2\pi}(1-\omega)^{2K-1}\left(2\pi\omega^2 - (2\pi + \lambda\gamma^2 + 2\lambda\eta)\omega + \lambda\gamma^2\right). \tag{B.5}$$

We can already see that 1 is an eigenvalue with multiplicity at least $2K - 1$. The other two eigenvalues are given by,

$$\omega_{\pm}(\eta) := \frac{1}{4\pi}\left(2\pi + \lambda\gamma^2 + 2\lambda\eta \pm \sqrt{(2\pi + \lambda\gamma^2 + 2\lambda\eta)^2 - 8\lambda\pi\gamma^2}\right). \tag{B.6}$$

Now since $1 \le \eta \le \pi^4/90$ there exists a constant $C > 0$ independent of $K$ such that,

$$\forall K \in \mathbb{N}: \quad \kappa(\widetilde{\mathbb{A}}(\lambda, c)) = \frac{\max\{1, \omega_+\}}{\min\{1, \omega_-\}} \le C. \tag{B.7}$$

Now suppose that one sets $\lambda = 2\pi/\gamma^2$. Then we find that

$$\omega_{\pm}(\eta) := 1 + \eta/\gamma^2 \pm \sqrt{2\eta/\gamma^2 + \eta^2/\gamma^4}, \tag{B.8}$$

and hence,

$$\kappa(\widetilde{\mathbb{A}}(\lambda, \gamma)) = \frac{1 + \sqrt{2\eta}/c + O(1/\gamma^2)}{1 - \sqrt{2\eta}/\gamma + O(1/\gamma^2)} = 1 + O(1/\gamma) \qquad \text{for } \gamma \to \infty. \tag{B.9}$$

As a result, $\lim_{\gamma \to +\infty} \kappa(\widetilde{\mathbb{A}}(2\pi/\gamma^2, \gamma)) = 1$. $\qquad\square$

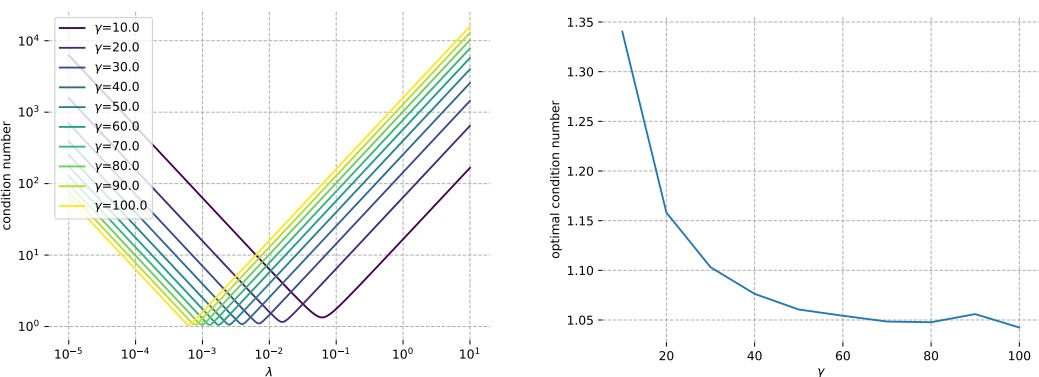

Figure 3: Evolution of condition number of preconditioned matrix for different $\gamma$ and $\lambda$ (left) and for different $\gamma$ for the optimal $\lambda$ (right).

## B.2 CHOOSING $\lambda$

As mentioned in the main text, it seems natural to suggest that the parameter $\lambda$ in loss (2.2) should be chosen as

$$\lambda^* := \min_{\lambda} \kappa(\mathbb{A}(\lambda)), \tag{B.10}$$

in order to obtain the smallest condition number of $\mathbb{A}$ and accelerate convergence, provided that it can be calculcated in a numerically stable way.

### B.2.1 POISSON EQUATION

As a first example, we revisit the setting of Theorem 3.1 where we considered solving the Poisson equation in one dimension by learning the coefficients of a basis of Fourier features (see formulas in main text).

In Figure 8, it was already shown how the condition number of $\mathbb{A}(\lambda)$ changes for various values of $\lambda$ and $K$. In particular, there is a very clear minimum in condition number for every $K$. In Figure 4 we can clearly see that the optimal $\lambda^*$ scales as $K^2$.

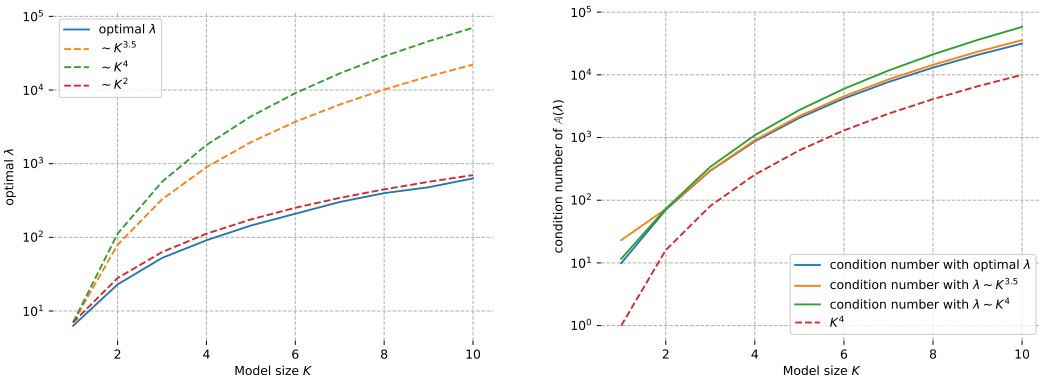

Figure 4: *Left:* optimal $\lambda$ in terms of $K$. *Right:* Evolution of condition number of $\mathbb{A}(\lambda)$ in terms of model size $K$ for multiple choices of $\lambda$.

We now compare our choice of $\lambda$ as in equation B.10 with related work. We first compare with Wang et al. (2021a). In this work, the authors propose a learning rate annealing algorithm to iteratively update $\lambda$ throughout training. At each iteration, they define

$$\lambda^* = \frac{\max_k \left| (\nabla_\theta R(\theta))_k \right|}{(2K+1)^{-1} \sum_k \left| (\nabla_\theta B(\theta))_k \right|}. \tag{B.11}$$

In our example it holds that $(\nabla_\theta R(\theta))_k = k^4 \theta_k$ for $k \neq \ell$ and $(\nabla_\theta R(\theta))_\ell = \ell^4(\theta_k - 1)$. We find that $\max_k \left| (\nabla_\theta R(\theta))_k \right| \sim K^4$ at initialization. Next we calculate that (given that at initialization it holds that $\theta_m \sim \mathcal{N}(0,1)$ iid),

$$\mathbb{E} \sum_k \left| (\nabla_\theta B(\theta))_k \right| = \sum_{k=0}^{K} \mathbb{E} \left| \sum_{m=0}^{K} \theta_m (-1)^{k+m} \right| = \sum_{k=0}^{K} \mathbb{E} \left| \sum_{m=0}^{K} \theta_m \right| = (K+1)^{3/2} \tag{B.12}$$

This brings us to the rough estimate that $(2K+1)^{-1} \sum_k \left| (\nabla_\theta B(\theta))_k \right| \sim \sqrt{K}$. So in total we find that $\lambda^* \sim K^{3.5}$.

Next, we compare with the proposed algorithm of Wang et al. (2022b). They propose to set

$$\lambda^* = \frac{Tr(K_{rr}(n))}{Tr(K_{uu}(n))} \approx \frac{\int_\Omega \nabla_\theta \mathcal{D} u_\theta^\top \nabla_\theta \mathcal{D} u_\theta}{\int_{\partial\Omega} \nabla_\theta u_\theta^\top \nabla_\theta u_\theta} = \frac{2\pi \sum_{k=1}^{K} k^4}{K+1} \sim K^4. \tag{B.13}$$

So, both works propose a choice of $\lambda$ that increases a lot faster in $K$ than the optimal choice in terms of the condition number ($\lambda^* \sim K^2$). In Figure 4 (right) we compare the obtained the condition numbers for the various choices of $\lambda^*$ and we observe that the relative difference is however small for small $K$ but increases for larger $K$. This phenomenon can be explained by the fact that $\kappa(\mathbb{A}(\lambda))$ has a relatively wide minimum for larger $K$.

### B.2.2 LINEAR ADVECTION EQUATION

As a second example, we revisit the linear advection equation that was studied in Section 3. The PDE is given by $u_t + \beta u_x = 0$ on $(-\pi, \pi)^2$ and we use periodic boundary conditions. As a model we

use $u_\theta(x,t) = \sum_{k\in\mathbb{Z}^2, \|k\|_\infty \leq K} \mathbf{e}_k(x,t)$, where for instance $\mathbf{e}_k(x) = \sin(k_1 x + k_2 t)$ when $k_1 < 0$ and $\mathbf{e}_k(x) = \cos(k_1 x + k_2 t)$ when $k_2 \geq 0$.

One can calculate that $\mathbb{A}_{km} = (k_2 + \beta k_1)^2 \delta_{km} + \lambda \delta_{k_1 m_1}$, where the boundary term now comes form the initial condition for $u_\theta(x,0)$, and where we rescaled $\mathbb{A}$ and $\lambda$ to get rid of any multiplicative factors stemming from the fact that we use unnormalized Fourier features. Re-indexing from a matrix $\mathbb{A}$ with indices in $\mathbb{Z}^2$ to a matrix $\mathbb{A}'$ with indices in $\mathbb{Z}$ such that $\mathbb{A}'_{Kk_1+k_2, Km_1+m_2} = \mathbb{A}_{km}$ then gives the formula $\mathbb{A}'(\lambda) = \mathrm{Id} \otimes C^2 + 2\beta C \otimes C + \beta^2 C^2 \otimes \mathrm{Id} + \lambda \mathrm{Id} \otimes (1 \cdot 1^\top)$, where $C$ is a diagonal matrix with $C_{\ell\ell} = \ell$.

In Figure 5 we plot the condition number of $\mathbb{A}'$ for various values of $\lambda$ and $\beta$. We see that the condition number increases with increasing $\beta$ and that for each $\beta$ there is a clear optimal $\lambda^*$. In Figure 6 we verify that $\lambda^*$ indeed scales as $\beta^2$, which was already observed in Figure 2.

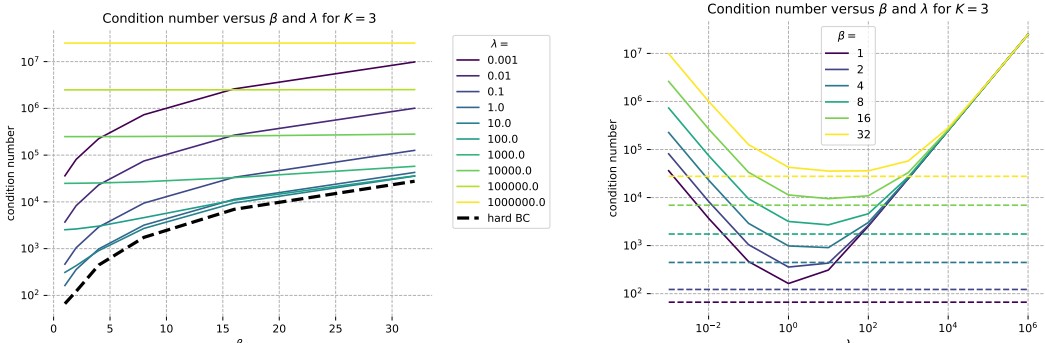

Figure 5: Evolution of condition number of $\mathbb{A}'$ for soft boundary conditions (full lines) and hard boundary conditions (dotted lines) in terms of $\beta$ and $\lambda$ for $K = 3$

This analysis also raises the questions what happens if $T$ is increased. This depends on whether the model size $K$ is changed along with $K$ or not.

- *$T$ is increased but the number of used Fourier basis functions is unchanged.* The first part of $\mathbb{A}'$, i.e. the one coming from $\mathcal{D}^*\mathcal{D}$, is independent of $\lambda$ and scales linear in $T$. As a result, rescaling $T$ corresponds to rescaling $\lambda$. Thus, the optimal $\lambda^*$ will change but the optimal condition number will not depend on $T$.

- *$T$ is increased and the number of Fourier basis functions is increased accordingly in time.* This essentially corresponds to changing $\beta$ and rescaling the measure. Moreover, this is intrinsically connected to what happens for **domain decomposition** in the time dimension,

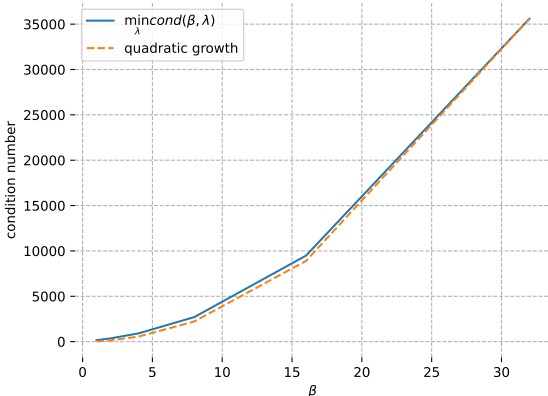

Figure 6: For each $\beta$ we have minimized the condition number in terms of $\lambda$ and displayed it. Growth is approximately quadratic for large $\beta$.

which on its own can be seen as a form of a **causal learning** strategy (Wang et al., 2022a). Since the condition number scales as $\beta^2$ one can argue that splitting the time domain in $N$ parts will reduce the condition number of each individual submodel by a factor of $N^2$.

### B.3 HARD BOUNDARY CONDITIONS

In the previous section, we have seen how varying the weighting parameter $\lambda$ of the boundary term can influence the condition number of $\mathbb{A}$. In view of Theorem 2.4 this is due to the fact that one changes the operator $\mathcal{A}$. In this section, we argue that different choices to implement *hard boundary conditions* can also change the condition number of $\mathbb{A}$, through changing the operator $TT^*$. We also give some examples where the condition number for hard boundary conditions is strictly smaller than for soft boundary conditions i.e., $\kappa(\mathbb{A}_{\text{hard BC}}) < \min_\lambda \kappa(\mathbb{A}_{\text{soft BC}}(\lambda))$.

**Demonstrative examples.** We first consider a toy model to demonstrate these phenomena in a straightforward way. We consider again the Poisson equation $-\Delta u = -\sin$ with zero Dirichlet boundary conditions. We choose as model $u_\theta(x) = \theta_{-1}\cos(x) + \theta_0 + \theta_1 \sin(x)$ and compare the condition number between various options.

- *Soft boundary conditions.* Using the approach discussed in Section B.2 we find that the condition number for the optimal $\lambda^*$ is given by $3 + 2\sqrt{2} \approx 5.83$.

- *Hard boundary conditions - variant 1.* A first common method to implement hard boundary conditions is to multiply the model with a function $\eta(x)$ so that the product exactly satisfies the boundary conditions, regardless of $u_\theta$. In our setting, we could consider $\eta(x)u_\theta(x)$ with $\eta(\pm\pi) = 0$. For $\eta = \sin$ the total model is given by

$$\eta(x)u_\theta(x) = -\frac{\theta_{-1}}{2}\cos(2x) + \frac{\theta_{-1}}{2} + \theta_0 \sin(x) + \frac{\theta_1}{2}\sin(2x), \qquad \text{(B.14)}$$

  and gives rise to a condition number of $4$. Different choices of $\eta$ will inevitably lead to different condition numbers.

- *Hard boundary conditions - variant 2.* Another option would be to subtract $u_\theta(\pi)$ from the model so that the boundary conditions are exactly satisfied. This corresponds to the model,

$$u_\theta(x) - u_\theta(\pi) = \theta_{-1}(\cos(x) + 1) + \theta_1 \sin(x) \qquad \text{(B.15)}$$

  Note that this implies that one can discard $\theta_0$ as parameter, leaving only two trainable parameters. The corresponding condition number is $1$.

**Linear advection equation.** With these instructive examples in place, we once again revisit the linear advection equation (as in **SM B.2.2**). If $u(x,0) = \sin(ax)$ then we could define our model as,

$$u_\theta(\mathbf{x}) = \sum_k \theta_k(\mathbf{e}_k(\mathbf{x}) - \mathbf{e}_k(x,0)) + \sin(ax), \qquad \text{(B.16)}$$

and correspondingly,

$$\mathcal{D}u_\theta(\mathbf{x}) = \sum_k \theta_k((k_2 + \beta k_1)\mathbf{e}_k(\mathbf{x}) - \beta k_1 \mathbf{e}_k(x,0)) + \beta a \cos(ax), \qquad \text{(B.17)}$$

and hence,

$$\mathbb{A}_{km} = (k_2 + \beta k_1)^2 \delta_{km} - \beta k_1(k_2 + \beta k_1)\delta_{k_1 m_1}[\delta_{k_2 0} + \delta_{m_2 0}] + (\beta k_1)^2 \delta_{k_1 m_1}. \qquad \text{(B.18)}$$

If we define the matrix $E = 1 \cdot e_0^T$ by $E_{ab} = \delta_{a0}$ then we can write the re-indexed matrix $\mathbb{A}'$ as,

$$\mathbb{A}' = \text{Id} \otimes C^2 + 2\beta C \otimes C + \beta^2 C^2 \otimes \text{Id} - \beta(\beta C \otimes C + \beta C^2 \otimes \text{Id})(\text{Id} \otimes (E + E^\top)) + C^2 \otimes (1 \cdot 1^\top). \qquad \text{(B.19)}$$

We find that for this specific case the condition number of $\mathbb{A}'$ with hard boundary conditions is always smaller than that with soft boundary conditions, for any $\beta$ or $\lambda$ (see Figure 5).

### B.4 SECOND-ORDER OPTIMIZERS

Second-order optimizers distinguish themselves from first-order optimizers, such as gradient descent, by using information of the Hessian in their update formula. In the case of **Newton's method**, the Hessian is used to precondition the gradient. We explore the connection of preconditioning based on our matrix $\mathbb{A}$ (2.6) with Newton's method.

Given any loss $\mathcal{J}(\theta)$, Newton's method's gradient flow is given by

$$\frac{d\theta(t)}{dt} = -\gamma H[\mathcal{J}(\theta(t))]^{-1}\nabla_\theta \mathcal{J}(\theta(t)), \tag{B.20}$$

where $H$ is the Hessian. In our case, we consider the physics-informed loss $L(\theta) = R(\theta) + \lambda B(\theta)$ (2.2) and consider the model $u_\theta(x) = \sum_\ell \theta_\ell \phi_\ell(x)$, which corresponds to a linear model or a neural network in the NTK regime (e.g. Lemma 2.2).

Using that $\partial_{\theta_i}\partial_{\theta_j}u_\theta = 0$, we calculate,

$$
\begin{aligned}
\partial_{\theta_i}\partial_{\theta_j}L(\theta) &= \int_\Omega (\mathcal{L}\partial_{\theta_i}u_\theta(x)) \cdot \mathcal{L}\partial_{\theta_j}u_\theta(x)dx + \lambda \int_{\partial\Omega} \partial_{\theta_i}u_{\theta(t)}(x) \cdot \partial_{\theta_j}u_\theta(x)dx \\
&= \int_\Omega \mathcal{D}\phi_i(x) \cdot \mathcal{D}\phi_j(x)dx + \lambda \int_{\partial\Omega} \phi_i(x) \cdot \phi_j(x)dx \\
&= \mathbb{A}_{ij}.
\end{aligned}
\tag{B.21}
$$

We conclude that $\mathbb{A}$ is identical to the Hessian of the loss. Hence, using Newton's method automatically leads to perfect preconditioning. However, the high potential cost of computing the exact Hessian inhibits the use of these class of optimizers in practice.

## C EXPERIMENTAL DETAILS AND ADDITIONAL RESULTS

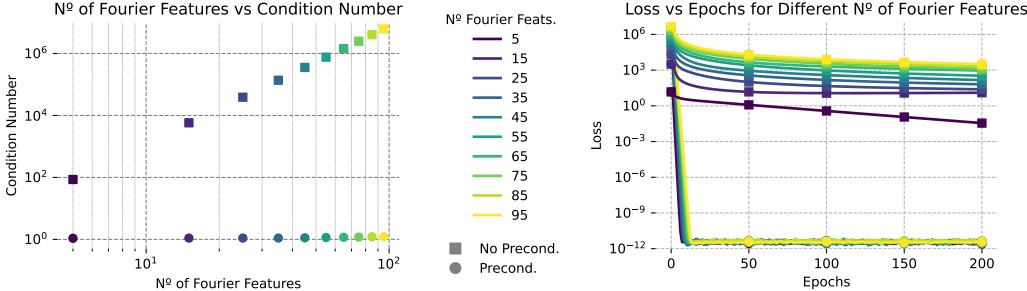

Figure 7: Helmholtz equation with Fourier features. *Left:* Optimal condition number vs. No. Fourier features. *Right:* Training curves for the unpreconditioned and preconditioned Fourier features.

Throughout all the experiments, we use `torch` version 2.0.1 and incorporate functional routines from `torch.func`. The matrix $\mathbb{A}$ is computed using the formula

$$\mathbb{A}_{i,j} = \langle \mathcal{D}\phi_i, \mathcal{D}\phi_j \rangle_{L^2(\Omega)} + \lambda \langle \phi_i, \phi_j \rangle_{L^2(\partial\Omega)}.$$

Here, $\phi_k := \partial_{\theta_k}u(\cdot; \theta_0)$ is initially obtained through autograd. Subsequently, the differential operator $\mathcal{D}$ is applied, also leveraging autodiff. For the scalar product, we employ Monte Carlo approximation and can utilize the same input coordinates used for training. In scenarios where the matrix becomes too large—often the case for large networks—numerical approximations of the $\mathcal{D}$ operator can be employed to reduce both computational and memory load.

For training, in order to avoid random errors we fix the grid to an equispaced grid of a size large enough to resolve the neural network/linear function.

Once the matrix $\mathbb{A}$ is computed, for the linear models we use a routine to compute its condition number and find the optimal $\lambda$ before training using the black-box gradient free 1d optimisation method from `scipy.optimize.golden`. The learning rate is then chosen as $1/\lambda_{\max}$. For models with MLPs this is no longer possible because of the zero eigenvalues that are plaguing the matrix; we resort to grid search sweeping wide ranges of learning rates and $\lambda$ values.

Figure 8: Sensitivity of the condition number on parameter $\lambda$, the multiplier between boundary and residual loss. In the non-preconditioned and preconditioned case (left and right, respectively), $\lambda$ has a non-negligible impact on the condition number. However, in the preconditioned case, this impact is always the same for different Fourier features, and thus plots for different Fourier features overlap. This implies that the optimal $\lambda$ will not have to be adjusted when changing this hyperparameter, as opposed to the non-preconditioned case.

### C.1 POISSON AND HELMHOLTZ, 1D

We solve the following boundary value problems on the domain $[-\pi, \pi]$. For the experiments with Fourier features the network is already periodic and thus there is no need to enforce these conditions.

**Poisson equation, 1d.** The solution is given by $u(x) = \sin(kx)$.

$$
\begin{aligned}
u''(x) &= -k^2 \sin(kx), \\
u(-\pi) &= 0,\ u(\pi) = 0.
\end{aligned}
\tag{C.1}
$$

**Helmholtz equation, 1d.** The solution is given by $u(x) = \cos(\omega x)$.

$$
\begin{aligned}
u''(x) + \omega^2 u(x) &= 0, \\
u(0) = 1,\ u'(0) &= 0, \\
u(-\pi) = u(\pi),\ u'(-\pi) &= u'(\pi).
\end{aligned}
\tag{C.2}
$$

For the Fourier features approximating the Helmholtz equation, we use exactly the same formulation as for the Poisson equation in the main text and a similar preconditioning matrix as in (3.3), but with diagonal entries scaling as $\frac{1}{|k^2 - \omega^2|}$. The results for Helmholtz equation, presented in Figure 7, are entirely analogous to the observations for the Poisson equation in Main text Figure 1. All the observations done on those two cases in the main paper and the appendices are also valid when looking at the mean squared error, see Figure 10 and 11. In Table 1 we report the computational time for one epoch in the preconditioned and unpreconditioned cases where, for fairness, we computed the preconditioning at each step instead of computing it once for all at the beginning of the training.

**Architectural Details.** For the MLP we used a 3 hidden neural network with a hidden dimension of $64$ and a hyperbolic tangent *tanh* activation function. All models are optimized using SGD.

### C.2 LINEAR ADVECTION EQUATION, 2D

We solve the following boundary value problems on the domain $\Omega := [0, 2\pi] \times [0, 1]$:

$$
\begin{aligned}
\frac{\partial u}{\partial t} + \beta \frac{\partial u}{\partial x} &= 0 \\
u(x, 0) = \sin(x), &\quad x \in [0, 2\pi] \\
u(0, t) = u(2\pi, t), &\quad t \in [0, 1]
\end{aligned}
\tag{C.3}
$$

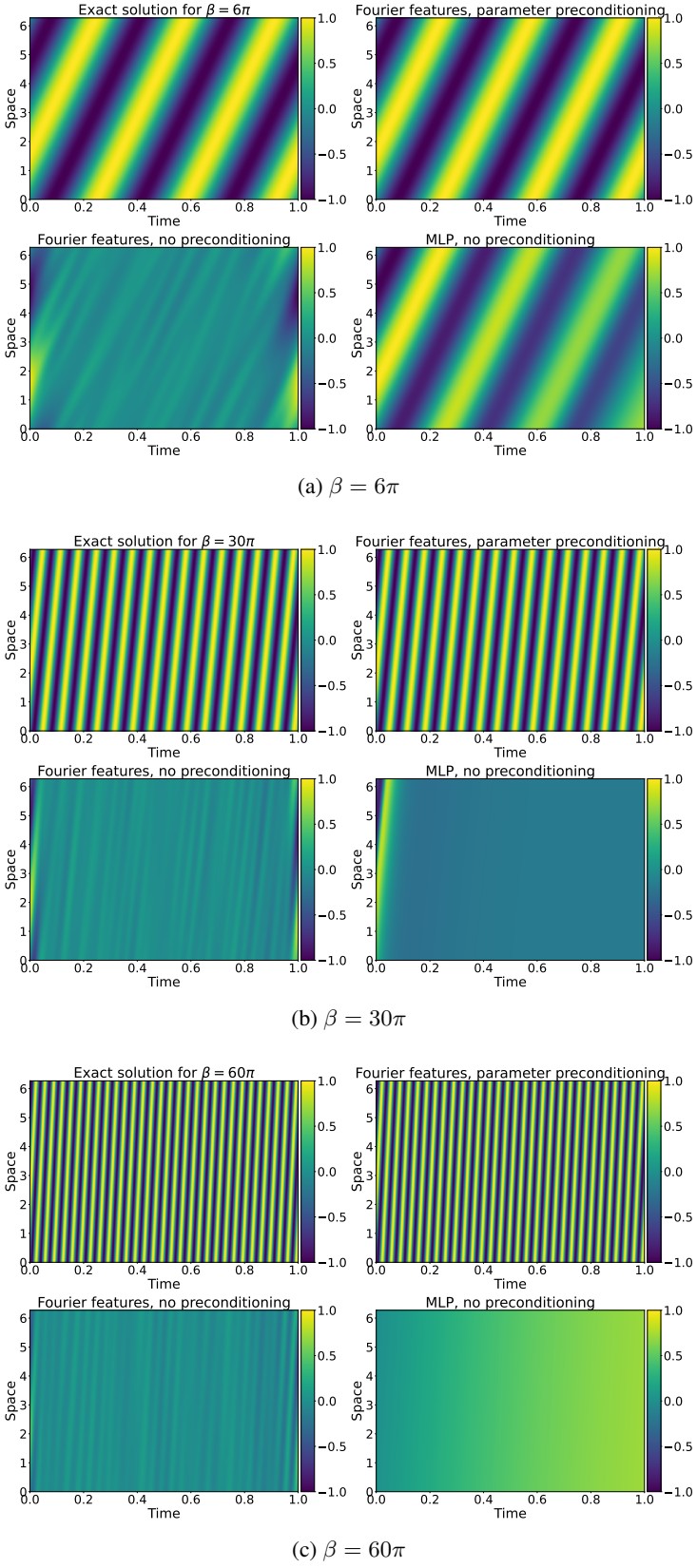

Figure 9: Solutions for the linear advection experiment

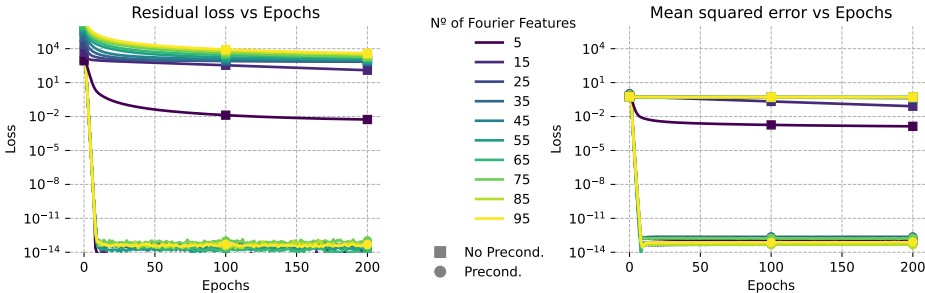

Figure 10: Training curves and mean squared error for the Fourier basis in the Poisson problem for different numbers of Fourier features using preconditioned SGD and Adam.

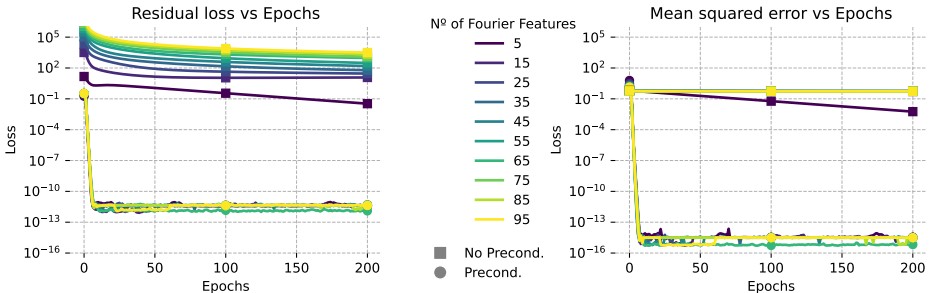

Figure 11: Training curves and mean squared error for the Fourier basis in the Helmholtz problem for different numbers of Fourier features using preconditioned SGD and Adam.

where $\beta$ is the speed of advection.

Each model $u_\theta$, with $\theta$ the set of its parameters, is trained with a Physics-Informed Loss defined as:

$$R(\theta) = \int_\Omega |\partial_t u_\theta + \beta \partial_x u_\theta|^2 \, dx dt$$

$$B(\theta) = \int_{[0,2\pi]} \left| u_\theta(x,0) - \sin(x) \right|^2 dx + \int_{[0,1]} \left| u_\theta(0,t) - u_\theta(2\pi,t) \right|^2 dt$$

$$L(\theta) = R(\theta) + \lambda B(\theta) \tag{C.4}$$

The solution of such differential equation is $(x,t) \mapsto \sin(x - \beta t)$.

**Architectural Details.** We test two models, a linear model via a two-dimensional Fourier series with bounded spectrum and a MLP with 5 layers, 50 neurons per hidden layer, and $\tanh$ activation function as used in Krishnapriyan et al. (2021). More explicitly, the ansatz for the linear model is:

$$u_\theta(x,t) = \frac{1}{\sqrt{\pi}} \sum_{k=-5}^{5} \sum_{m=0}^{30} \left( \theta_{k,m}^{\cos} \cos(kx + 2\pi mt) + \theta_{k,m}^{\sin} \sin(kx + 2\pi mt) \right) \tag{C.5}$$

with $\theta := \left\{ \theta_{k,m}^{\cos}, \theta_{k,m}^{\sin} \right\}_{k,m}$.

In the preconditioned case, the preconditioning matrix is set to a diagonal matrix with diagonal terms $\frac{1}{|m+\beta k|}$ for $(k,m) \neq (0,0)$. For the parameter $\theta_{0,0}^{\cos}$ we set the preconditioning factor to 1.

**Training details.** We discretize the domain $\Omega$ on a grid of size $256 \times 100$ for the learning process to be consistent with the configuration proposed in Krishnapriyan et al. (2021) and a grid of size $256 \times 2048$ to compute the matrix $\mathbb{A}$ defined in 2.6 with finite differences.

For the linear model, we use classical batch gradient descent on the full grid with and without preconditioning for 200 epochs. The learning rate and $\lambda$ are automatically defined to minimize the

Table 1: Computational time per step in milliseconds. The preconditioning time refers to the parameters preconditioning of the Fourier basis via the approximate analytical conditioning described in the Appendices.

| Problem | Preconditioning | SGD |
|---|---|---|
| Poisson 1D | $14.1 \pm 2.1$ | $8.9 \pm 2.2$ |
| Helmholtz 1D | $15.9 \pm 3.2$ | $11.2 \pm 2.8$ |
| Advection 2D | $17.2 \pm 1.5$ | $10.9 \pm 0.7$ |

condition number of $\mathbb{A}$ via a Golden section method. For the MLP, we use Adam on the full grid without preconditioning for 10000 epochs. The learning rate is set to 0.0001 and $\lambda$ is set to 1 via grid search as we encountered out-of-memory errors when computing $\mathbb{A}$. Training curves for the linear model with and without conditioning for different $\beta$ and a plot of the condition number with respect to $\beta$ is given in Figure 2. Training curves for the MLP for different $\beta$ are given in Figure 13.

**Results.** The behavior of condition numbers and loss functions for the linear model and its preconditioned version were already shown in Figure 2 and discussed in the main text. The loss functions with MLPs (trained with ADAM), for different $\beta$'s are shown in Figure 13 and we observe that the MLP (trained with ADAM) converged very slowly to an unacceptably large loss function of amplitude $10^{-2}$, which can be contrasted with the very fast decay of the loss function to $10^{-10}$ for the preconditioned Fourier model (Figure 2 (right)). These issues in training severely impact the overall quality of the results with the three models considered (unpreconditioned Fourier, preconditioned Fourier and MLP). In Figure 9, we plot the exact solution (in space-time) and compared it with the solutions obtained with the three models, at three different advection speeds $\beta = 6\pi, 30\pi, 60\pi$. We see from this figure that the unpreconditioned Fourier model fails completely, even at the slowest considered speed $\beta = 6\pi$. MLP (trained with ADAM) is better at the slowest speed than the unpreconditioned Fourier model but fails completely at the higher speeds, as already observed in Krishnapriyan et al. (2021). On the other hand, consistent with our theory and the low values of the loss function observed in Figure 2 (right), the preconditioned Fourier model was able to approximate the solution (in space-time) with high-accuracy, even for the fastest considered advection speed of $\beta = 60\pi$. This example clearly demonstrates the superiority of the preconditioned linear models over their unconditioned versions as well as over nonlinear neural network models. All of this observations are also valid when looking at the mean squared error (computed at the same collocation points as the training loss), see Figure 12. In Table 1 the computational time is reported for each method where, for fairness, we computed the preconditioning at each step instead of computing it once for all at the beginning of the training.

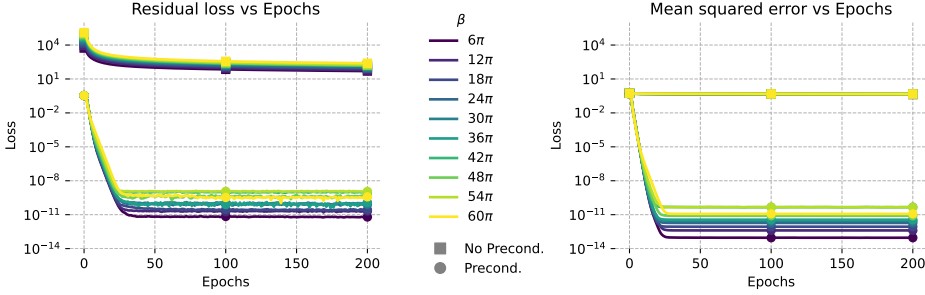

Figure 12: Training curves and mean squared error for the Fourier basis in the advection problem for different $\beta$ using preconditioned SGD and Adam.

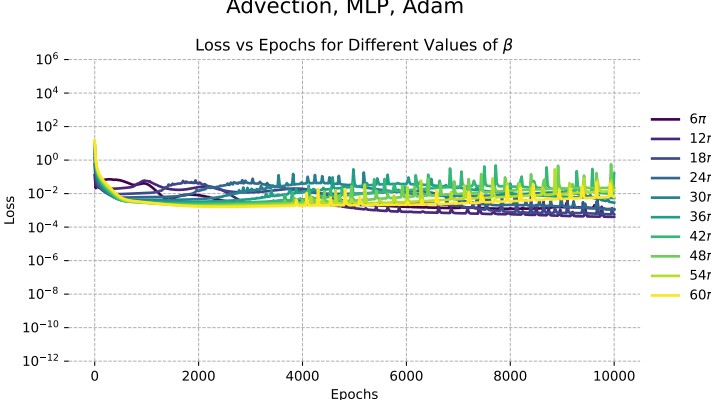

Figure 13: Training curves for the MLP in the advection problem for different $\beta$ using Adam.

## C.3 Preconditioning via Direct Inversion of the Matrix $\mathbb{A}$

In this section, we wish to validate the theory and assess the assumptions made in a practical, non-linear scenario. To this extent, we propose to directly precondition the gradient using the inverse of the matrix $\mathbb{A}$ defined in (2.6), recalled below:

$$\mathbb{A}_{i,j} = \langle \mathcal{D}\phi_i, \mathcal{D}\phi_j \rangle_{L^2(\Omega)} + \lambda \langle \phi_i, \phi_j \rangle_{L^2(\partial\Omega)}, \tag{C.6}$$

where $\phi_i(x) = \partial_{\theta_i} u(x; \theta_0)$. Given that the simplified gradient assumption holds, i.e. the smallness of the error term $\varepsilon$ in (2.7), (non-)convergence is entirely governed by the conditioning of the matrix $\mathbb{A}$. This matrix can be readily computed using autodifferentiation, first differentiating w.r.t. $\theta$, then w.r.t. the coordinates for the differential operator $\mathcal{D}$. The scalar products are then computed using a Monte Carlo approximation on the training points.

We then precondition the gradients, using the regularized inverse $\mathbb{A}_\varepsilon^{-1} := (\mathbb{A} + \varepsilon I)^{-1}$, the parameter updates given by:

$$\widehat{\theta}_{k+1} = \widehat{\theta}_k - \eta \mathbb{A}_\varepsilon^{-1} \nabla_\theta L(\widehat{\theta}_k). \tag{C.7}$$

Recalling equations (3.2) and (3.1), the conditioning of the problem now is governed by the matrix $\widetilde{\mathbb{A}} := \mathbb{A}_\varepsilon^{-\frac{1}{2}} \mathbb{A} \mathbb{A}_\varepsilon^{-\frac{1}{2}} \approx I$, whose condition number is thus approximately optimal. Experiments are conducted using a one-hidden layer, 32 neurons, $\tanh$ MLP, using a regularization parameter $\varepsilon \in [0.0001, 0.004]$, and double precision.

The preconditioned gradient is directly numerically computed via the resolution of a least-square problem of the form:

$$\underset{x}{\arg\min} \|\mathbb{A}_\varepsilon x - \nabla_\theta L(\widehat{\theta}_k)\|_2. \tag{C.8}$$

We must underline that the regularizing term $\varepsilon I$ and the double precision are crucial for numerical stability as the matrix $\mathbb{A}$ is highly ill-conditioned for such MLP. Also, we computed the matrix $\mathbb{A}$ for each new set of parameters, solving the least-square problem at each epoch instead of solving it only at the initialization.

For Poisson, Helmholtz, and Advection, we compare the convergence of the preconditioned MLP to a non-preconditioned MLP, trained with SGD and Adam respectively. In figures 14, 15, 16, we report training curves (left) as well as mean squared errors (right) for different PDE parameters. We observe that preconditioning with this strategy not only consistently yields much lower errors, but also much faster training times. Solutions for the advection equation can be visually inspected in Figures 18 and 19: consistent with findings in Krishnapriyan et al. (2021), the MLP trained

with no preconditioning does not converge to the true solution, even after an indefinite number of timesteps. In contrast, solutions with the preconditioned MLP are visually indistinguishable from the true solution. However, this preconditioning strategy comes at a price, as computational costs are higher (as opposed to the preconditioning strategy described in the main paper), as reported in Table 2.

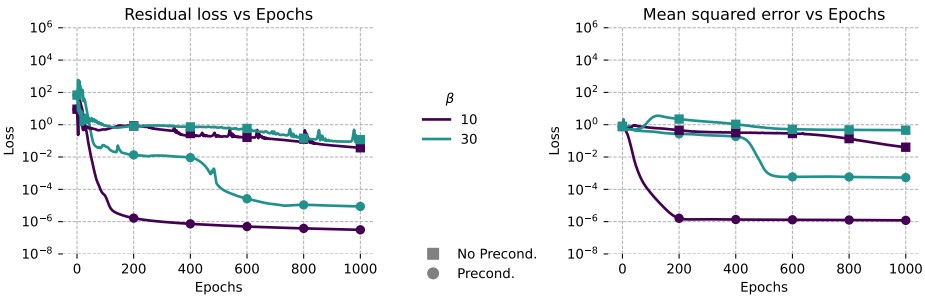

Figure 14: Training curves and mean squared error for the shallow MLP in the advection problem for different $\beta$ using preconditioned SGD and Adam.

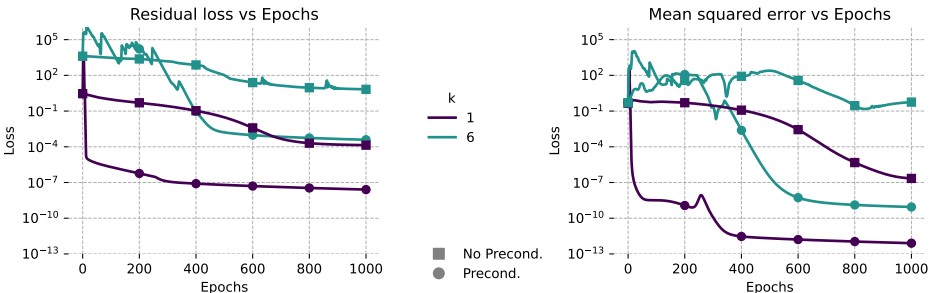

Figure 15: Training curves and mean squared error for the shallow MLP in the Poisson problem for different $k$ using preconditioned SGD and Adam.

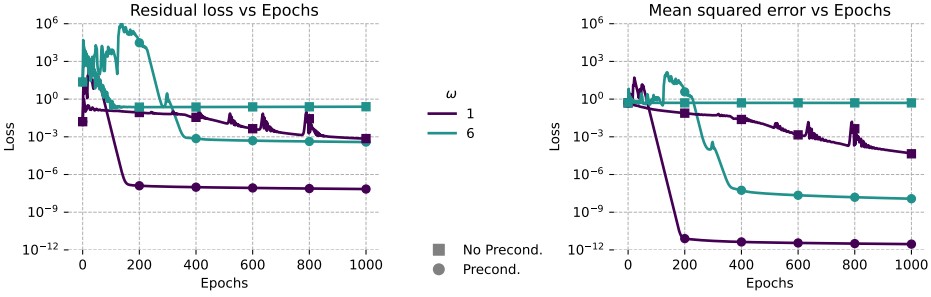

Figure 16: Training curves and mean squared error for the shallow MLP in the Helmholtz problem for different $\omega$ using preconditioned SGD and Adam.

### C.4 PRECONDITIONING NONLINEAR PHYSICS-INFORMED MACHINE LEARNING MODELS

We investigate conditioning of nonlinear models by considering the Poisson equation on $(-\pi, \pi)$ and learning its solution with neural networks of the form $u_\theta(x) = \Phi_\theta(x)$, with $x \in (-\pi, \pi)$ and $\Phi_\theta$ being a three hidden layer MLP. We compute the resulting matrix $\mathbb{A}$ (2.6) and plot its normalized eigenvalues in Figure 20 (left) to observe that most of the eigenvalues are clustered near zero, in accordance with the fact that Hessians (which are correlated with $\mathbb{A}$) for neural networks have a large number of (near) zero eigenvalues (Ghorbani et al., 2019). Consequently, it is difficult to

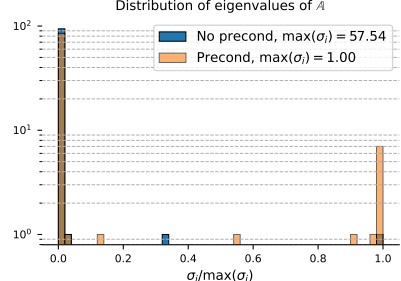 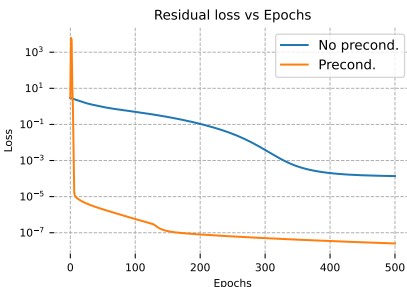

Figure 17: Left: Normalized Spectrum of the shallow MLP and preconditioned shallow MLP for the Poisson Equation. Right: Training loss for the shallow MLP and preconditioned shallow MLP for the Poisson Equation.

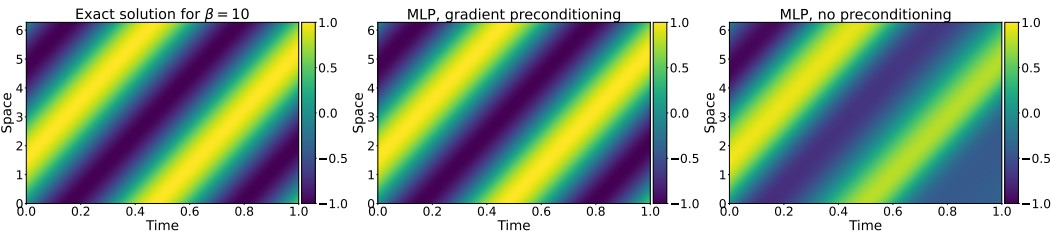

Figure 18: Comparison of the solutions for the shallow MLP in the advection problem using preconditioned SGD and Adam with $\beta = 10$.

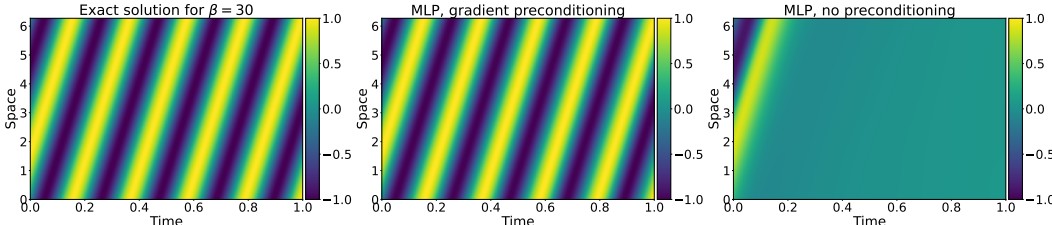

Figure 19: Comparison of the solutions for the shallow MLP in the advection problem using preconditioned SGD and Adam with $\beta = 30$.

Table 2: Computational time per step in milliseconds. The preconditioning time refers to the gradient preconditioning of the shallow MLP via inversing the $\mathbb{A}$ matrix.

| Problem | Preconditioning | Adam |
|---------|-----------------|------|
| Poisson 1D | $16.2 \pm 2.0$ | $8.2 \pm 1.8$ |
| Helmholtz 1D | $20.5 \pm 3.6$ | $9.7 \pm 2.0$ |
| Advection 2D | $200.0 \pm 25.8$ | $5.9 \pm 1.1$ |

analyze the condition number per se. However, we also observe from this figure that there are only a couple of large non-zero eigenvalues of $\mathbb{A}$ indicating a very uneven spread of the spectrum. It is well-known in classical numerical analysis (Trefethen & Embree, 2005) that such spread-out spectra are very poorly conditioned and this will impede training with gradient descent. This is corroborated in Figure 20 (right) where the physics-informed MLP trains very slowly. Moreover, it turns out that preconditioning also localizes the spectrum (Trefethen & Embree, 2005). This is attested in Figure 20 (left) where we see the localized spectrum of the preconditioned Fourier features model considered previously, which is also correlated with its fast training (Figure 1 (right)).

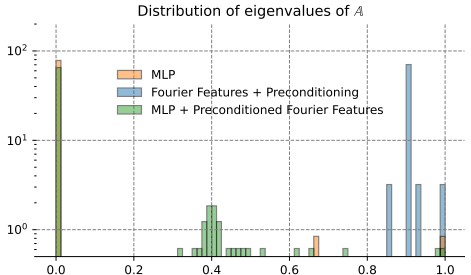 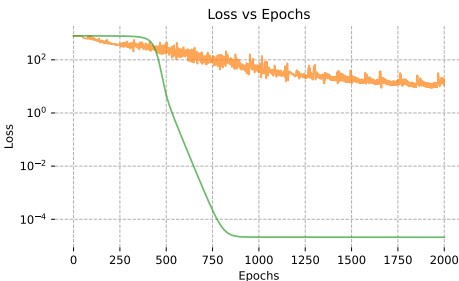

Figure 20: Poisson equation with MLPs. *Left:* Histogram of normalized spectrum (eigenvalues multiplied with learning rate). *Right:* Loss vs. number of epochs.

However, preconditioning $\mathbb{A}$ for nonlinear models such as neural networks can, in general, be hard. Here, we consider a simple strategy by coupling the MLP with Fourier features $\phi_k$ defined above, i.e., setting $u_\theta = \Phi_\theta \left( \sum_k \alpha_k \phi_k \right)$. Intuitively, one must carefully choose $\alpha_k$ to control $\frac{d^{2n}}{dx^{2n}} u_\theta(x)$ as it will include terms such as $(\sum_{k \neq 0} \alpha_k (-k)^{2n} \phi_k) \Phi_\theta^{(2n)} \left( \sum_k \alpha_k \phi_k \right)$. Hence, such rescaling can better condition the Hermitian square of the differential operator. To test this for Poisson's equation, we choose $\alpha_k = 1/k^2$ (for $k \neq 0$) in this FF-MLP model and present the eigenvalues in Figure 20 (left) to observe that although there are still quite a few (near)-zero eigenvalues, the number of non-zero eigenvalues is significantly increased leading to a much more even spread in the spectrum, when compared to the unpreconditioned MLP case. This possibly accounts for the fact that the resulting loss function decays much more rapidly, as shown in Figure 20 (right), when compared to unpreconditioned MLP.

