# OpenReview forum: "An operator preconditioning perspective on training in physics-informed machine learning"
_ICLR.cc/2024/Conference — ICLR 2024 poster_

### Official Review · Reviewer_Wj3M · 2023-10-29

**Soundness:** 4 excellent
**Presentation:** 2 fair
**Contribution:** 3 good
**Rating:** 5
**Confidence:** 5

**Summary:**

The paper mainly explores how the convergence of training in Physics-Informed Neural Networks (PINNs) is dependent on the condition number of the operators constituting the Partial Differential Equations (PDEs). From the perspective of novelty, the idea posited in this paper is quite interesting. In terms of conclusions, the paper articulately presents the theoretical result that a lower condition number leads to better convergence of PINNs. Moreover, a preconditioning method for linear PINNs is proposed in the paper, significantly reducing the condition number and accelerating the training process.

Overall, I appreciate the theoretical analysis conducted in this paper, which I believe meets the submission standards for ICLR. However, the experimental section is somewhat weak (as noted in the drawbacks), and I believe that more extensive experiments should be conducted to validate the theoretical analysis presented in the paper. Additionally, some related works concerning hard constraints were overlooked, specifically the following two papers:

**Reference**
1. A Unified Hard-Constraint Framework for Solving Geometrically Complex PDEs (https://arxiv.org/abs/2210.03526)
2. PFNN: A Penalty-Free Neural Network Method for Solving a Class of Second-Order Boundary-Value Problems on Complex Geometries (https://arxiv.org/abs/2004.06490)

**Strengths:**

1. The idea presented in this paper is novel, to the best of my knowledge.
2. The paper's approach of using condition number analysis to study the convergence of PINNs is innovative, providing fundamental theoretical guarantees for PINN convergence.
3. The writing of most sections is easy to understand. Overall, the theoretical analysis part of this paper is very engaging to read.

**Weaknesses:**

1. The experimental section of the paper is quite weak. It solely provides a method for reducing the condition number in linear PDEs using PINNs based on the theoretical analysis presented, which considerably narrows the applicability of the proposed method.
2. Some parts of the paper are not clearly written, such as the calculation of the preconditioner matrix P, and the discussion regarding complexity appears to be vague.
3. The logic in the latter experimental and discussion sections is somewhat jumbled. For instance, the last two sections, in my opinion, should be consolidated into one section as related work.

**Questions:**

None.

---

> ### Author Response · Authors · 2023-11-18
> **Reply to Reviewer Wj3M (part 1)**
>
> We thank the reviewer for appreciating the theoretical contributions of our paper. Below, we address the questions raised by the reviewer and thank the reviewer in advance for their patience in reading our reply.
> 1. (W1) Regarding the reviewer's concern that the **The experimental section of the paper is quite weak**, we would like to, at length, frame the contributions of our paper. To this end, we take the liberty to repeat the relevant portion of our reply to all the reviewers above here: At the outset, it is amply clear in the literature that a major issue with physics-informed machine learning models such as PINNs is the difficulty of training them. This has been widely recognised and several ad-hoc strategies have been proposed to improve training. However, it is also clear that a suitable theoretical framework that precisely formulates the problem of optimizing physics-informed loss functions was missing. We believe that a lack of precision in defining what the problem actually is also impedes finding suitable measures for ameliorating the problem itself.
>
>     With this background in mind, our main aim in the paper was precisely to propose a suitable theoretical framework in which the issues affecting training of physics-informed machine learning models can be formulated and analyzed. To this end, we have rigorously connected the rate of convergence of gradient descent algorithms for physics-informed loss functions to the condition number of an operator, which involves the Hermitian-square of the underlying differential operator. This contribution is novel to the best of our knowledge and has been acknowledged as such by most of the reviewers. A corollary of this observation is the fact that preconditioning (now defined rigorously as reducing the condition number of the underlying operator) is necessary to accelerate training of physics-informed machine learning models. Again, this provides a mathematically precise definition of preconditioning for these models and stands in stark contrast to the existing literature where the words *preconditioning* and *condition numbers* are used very loosely and imprecisely. These two aforementioned aspects constitute our main contribution and has been emphasised as such repeatedly in our paper.
>
>     Given the precise notion of preconditioning that we propose, it was incumbent upon us to also analyse several existing strategies for improving training in physics-informed machine learning through the perspective of our framework of operator preconditioning. To this end, we have in Section 3 and in the **SM**, analyzed via a combination of theoretical analysis and empirical evaluation, several such strategies (Choice of balance parameter $\lambda$, Hard Boundary conditions, Second-order optimizers, Domain decomposition and Casual learning, we will also include *Sobolev Training*) through the lens of operator preconditioning, providing a novel interpretation of these strategies and identifying conditions under which they might or might not work. We had gone further and analyzed an additional strategy that essentially boils down to *rescaling Fourier features*, whether with respect to linear basis functions or nonlinear neural networks. These are secondary contributions and we feel that it is not fair to judge the article based on secondary, rather than primary contributions.
>
>     Moreover, the fact that **only linear PDEs was considered** in the empirical evaluation was based on the observation that if physics-informed machine learning models do not work for simple linear PDEs, there is little hope of them working for complicated nonlinear PDEs. Furthermore, we follow widely cited papers such as Krishnapriyan et al, 2021, which flagged the problems of training PINNs by focussing considerably on simple linear PDEs. Finally, the page limit also automatically limits the scope of an article. Given these considerations, we believe that linear PDEs already provide a reasonably representative arena for analyzing the issue of training PINNs and related models and lessons learnt there can be applied to nonlinear PDEs in future works. In particular, at some level, nonlinear PDEs have to be **linearized** and the lessons learnt from linear PDEs will be germane. Finally in this context, we would like to mention that there is a tradition of analyzing training in machine learning through linear models (see ArXiv:1810.02281 and references therein) as analyzing gradient descent even with linear models is highly non-trivial and representative of the difficulties faced in training non-linear models.

---

> ### Author Response · Authors · 2023-11-18
> **Reply to Reviewer Wj3M (part 2)**
>
> 1. (continued) Finally, we have added a new section **SM** Section C.3, where we consider another general simple preconditioning strategy which consists of using an approximate inverse of the matrix $\mathbb{A}$ (Eqn. (2.6)) as a preconditioner for the gradient descent step, i.e., pre-multiplying $(\mathbb{A} + \varepsilon I)^{-1}$ to the gradient descent update. This provides an additional general preconditioning strategy and we believe further strengthens the empirical aspects of our paper.
> 2. (W2+W3) *Some parts of the paper are not clearly written* and *the last two sections, in my opinion, should be consolidated into one section as related work*. We thank the reviewer for this excellent suggestion. In the process of incorporating all the 6 reviewers' multiple suggestions, we will consider a restructuring of these sections with your suggestion in mind in a **CRV**, if accepted.
> 3. We thank the reviewer for pointing out additional relevant references for hard BCs and will incorporate them in a **CRV**, if accepted.
>
> We sincerely hope that we have addressed all the reviewer's concerns, particularly about the scope of empirical vs. theoretical contributions in our paper, to your satisfaction. We kindly request the reviewer to upgrade your assessment of our paper accordingly.

---

> > ### Author Response · Authors · 2023-11-21
> > **Requesting the reviewer for feedback**
> >
> > Due to the imminent closure of the discussion period, We kindly request the reviewer to provide us some feedback on our rebuttal. We are at your disposal for any further questions/clarifications regarding the paper or the rebuttal.

---

### Official Review · Reviewer_E6qc · 2023-10-30

**Soundness:** 2 fair
**Presentation:** 2 fair
**Contribution:** 2 fair
**Rating:** 6
**Confidence:** 4

**Summary:**

This paper studies the training dynamics of gradient descent (GD) for solving partial differential equations (PDE) with physics-informed neural networks (PINN). Theoretically, the authors analyze a linearized form of the GD update. Let $\mathcal{D}$ denote the differential operator associated with the underlying PDE. The authors show that the convergence rate of GD depends on the conditional number of the Hermitian square operator $\mathcal{D}^\ast\mathcal{D}$, which implies that preconditioning the operator $\mathcal{D}^\ast\mathcal{D}$ helps accelerate the training process. Empirically, the authors propose preconditioning strategies for several PDEs and provide numerical experiments to exhibit the effectiveness of using preconditioning during training.

**Strengths:**

This paper includes both theoretical analysis and numerical experiments to support the argument that preconditioning is essential for training PINNs to solve PDEs. For numerical studies, the authors provide plots of both the condition number and loss function to exhibit that preconditioning does improve the training efficiency. Moreover, the authors also explain why preconditioning serves as a general framework for improving training efficiency by showing that several training strategies, such as tuning the regularization parameter, enforcing hard boundary conditions, using high-order optimizers and domain decomposition, can be treated as preconditioning the operator in some way.

**Weaknesses:**

(1) The reviewer's first concern is that this paper doesn't discuss its relation to previous work on preconditioning. Specifically, one line of work that researchers within the ML for PDE community have studied is "Sobolev Training", where the main idea is to incorporate the Sobolev gradients of the training objective into the loss function. Examples of theoretical and empirical work in this direction include [1] (theory) and [2, 3, 4] (empirical studies). In fact, sobolev training can be interpreted as a form of operator preconditioning, which we will explain it briefly below: Consider the task of solving the PDE $\mathcal{D}u = f$ over the domain $\Omega$ with zero boundary condition via PINN, where $\mathcal{D}$ is some differential operator. Let $E(u) :=\frac{1}{2} \|\mathcal{D}u - f\|_{L^2(\Omega)}^2$ be the standard $L^2$ loss function, whose associated gradient flow is $u_t =  -\frac{\delta}{\delta u}E(u) = -\mathcal{D}^\ast(\mathcal{D}u-f)$.

In contrast, if we consider solving the same problem under the $H^2$ loss $E_{S}(u) := \frac{1}{2} \|(I-\Delta)(\mathcal{D}u - f)\|_{L^2(\Omega)}^2$  (Sobolev training), we have that new gradient flow is given by $u_t =  -\frac{\delta}{\delta u}E_S(u) = -\mathcal{D}^\ast\mathcal{P}^\ast\mathcal{P}(\mathcal{D}u-f)$, where the differential operator $\mathcal{P} = I - \Delta$. This implies that training under a loss function that includes Sobolev gradient is equivalent to preconditioning the Hermitian operator $\mathcal{D}^\ast\mathcal{D}$ via the differential operator $\mathcal{P}$. Another approach for reducing the condition number is to reformulate the problem of minimizing the $L^2$ loss as a min max optimization problem, which was proposed in recent work [5]

Moreover, the idea of using preconditioning/sobolev training to speed up optimization/training of a prespecified model in machine learning and scientific computing is not new either. A few examples of previous work include using preconditioning in neural network training [6, 7], application of Sobolev preconditioned GD in image processing [8, 9, 10] and graphics [11, 12], etc. Hence, the authors are encouraged to have a subsection discussing how this paper is related to previous work on the usage of preconditioning/sobolev training in ML for PDE and related fields.

(2) Secondly, given that previous work [1, 2, 3, 4, 5] have already studied how sobolev training, which is equivalent to preconditioning, accelerates the optimization process of solving PDEs with PINNs, the reviewer thinks that this paper's novelty seems to be a bit limited.
It seems to me that sobolev training is a natural and simple method to implement (The main idea is to use integration by parts to kill any term involving the derivatives of the parametrized PDE solution $u_{\theta}$). However, for the method proposed in this work, it seems hard to find a good preconditioner $P$ to reduce $\kappa(A)$, especially when the differential operator $\mathcal{D}$ and the neural network model $u_{\theta}$ becomes a bit complicated. Consequently, to complement the numerical experiments exhibited in this paper, it would probably be meaningful to test the proposed method's performance on more complicated nonlinear PDEs, such as the Allen-Cahn equation, the Schrodinger equation and the Navier Stokes equation used in [13].

(3) Thirdly, the presentation of this paper can possibly be improved from a few aspects: Firstly, the review of ML-based/Physics-Informed PDE solvers in the second paragraph of Section 1 seems to be incomplete. A lot of recent work in this field seems to be missing. The authors are encouraged to take a look at the first part of Subsection 1.1 in [1] for a complete review of related work in this field. Secondly, it's probably better to move the subsection of related work from Section 4 to Section 1 (Introduction), as this might help the readers under the main contributions presented at the end of the introduction in a better way. It would also be helpful to include a subsection of the paper's organization and a subsection discussing the mathematical notations used in this paper within Section 1 (Introduction). Moreover, there seem to be some typos in the proofs presented in this paper. A non-exhaustive list of typos is given below:

1. In (2.5), a factor of $\frac{1}{2}$ is missing in front of the Hessian term $H_k$. Similarly, the $\frac{1}{2}$ factor is also missing in the expansion of the two gradients $\nabla_{\theta}R(\theta_k)$, $\nabla_{\theta}B(\theta_k)$ and the error term $\epsilon_k$ in Appendix A.1.

2. Based on the definition of the vector $C$, the sign in front of $C$ in (2.7), (2.8) should be negative, i.e, $C$ should be replaced by $-C$. Also, the expression of $\theta$ in Theorem 2.3 should be $\theta = \theta_0 - A^{-1}C$. Similarly, $\eta \epsilon_k$ should be replaced by $-\eta \epsilon_k$ in (2.7)

3. In the proof of Lemma 2.1 presented in Appendix A.2, the term $\epsilon_k$ should be replaced by $\eta \epsilon_k$ in the first line (A.2) of the proof. Therefore, the proof and the main result of Lemma 2.1 also need to be slightly modified.

**Questions:**

1. Would it be possible for the authors to discuss the main difference between the preconditioning method proposed in this paper and sobolev training? Is there any numerical example that your method outperforms (higher accuracy, lower computational cost, etc) sobolev training when the models used for parametrizing the PDE solution are the same?

2.  The authors mentioned that there are empirical studies showing that first-order method are less effective compared to second order methods for physics-informed ML. Would it be possible for the authors to include related references here? It seems to me that adaptive methods like Adam can also improve the conditioning as it takes use of diagonal rescaling. Hence, it might be meaningful for the authors to provide some numerical examples to explain why preconditioned GD can outperform Adam/SGD.

A full list of references mentioned in this review is given below:

[1] Lu, Y., Blanchet, J., & Ying, L. (2022). Sobolev acceleration and statistical optimality for learning elliptic equations via gradient descent. Advances in Neural Information Processing Systems, 35, 33233-33247.

[2] Yu, J., Lu, L., Meng, X., & Karniadakis, G. E. (2022). Gradient-enhanced physics-informed neural networks for forward and inverse PDE problems. Computer Methods in Applied Mechanics and Engineering, 393, 114823.

[3] Son, H., Jang, J. W., Han, W. J., & Hwang, H. J. (2020). Sobolev training for the neural network solutions of pdes.

[4] Son, H., Jang, J. W., Han, W. J., & Hwang, H. J. (2021). Sobolev training for physics informed neural networks. arXiv preprint arXiv:2101.08932.

[5] Zeng, Q., Kothari, Y., Bryngelson, S. H., & Schäfer, F. (2022). Competitive physics informed networks. arXiv preprint arXiv:2204.11144.

[6] Czarnecki, W. M., Osindero, S., Jaderberg, M., Swirszcz, G., & Pascanu, R. (2017). Sobolev training for neural networks. Advances in neural information processing systems, 30.

[7] Amari, S. I., Ba, J., Grosse, R., Li, X., Nitanda, A., Suzuki, T., ... & Xu, J. (2020). When does preconditioning help or hurt generalization?. arXiv preprint arXiv:2006.10732.

[8] Zhu, B., Hu, J., Lou, Y., & Yang, Y. (2021). Implicit regularization effects of the Sobolev norms in image processing. arXiv preprint arXiv:2109.06255.

[9] Calder, J., Mansouri, A., & Yezzi, A. (2010). Image sharpening via Sobolev gradient flows. SIAM Journal on Imaging Sciences, 3(4), 981-1014.

[10] Richardson Jr, W. B. (2008). Sobolev gradient preconditioning for image‐processing PDEs. Communications in Numerical Methods in Engineering, 24(6), 493-504.

[11] Soliman, Y., Chern, A., Diamanti, O., Knöppel, F., Pinkall, U., & Schröder, P. (2021). Constrained willmore surfaces. ACM Transactions on Graphics (TOG), 40(4), 1-17.

[12] Yu, C., Brakensiek, C., Schumacher, H., & Crane, K. (2021). Repulsive surfaces. arXiv preprint arXiv:2107.01664.

[13] Raissi, M., Perdikaris, P., & Karniadakis, G. E. (2019). Physics-informed neural networks: A deep learning framework for solving forward and inverse problems involving nonlinear partial differential equations. Journal of Computational physics, 378, 686-707.

---

> ### Author Response · Authors · 2023-11-18
> **Response to Reviewer E6qc (part 1)**
>
> We thank the reviewer for their detailed feedback on our paper. Below, we address the questions raised by the reviewer and thank the reviewer in advance for their patience in reading our reply.
>
> 1. (W1): The reviewer has pointed to **Missing related work**. We would like to thank the reviewer for identifying very interesting related work, especially on Sobolev training and competitive PINNs that we will certainly include in a **CRV**, if accepted. Although we have taken care to mention previous work on preconditioning in the field of numerical analysis, we apologize having missed the stated work in ML and will include them in a **CRV**, if accepted.
> 2. (W2). The reviewer has addressed the issue of **Sobolev Training** and its relation to preconditioning at great length. We apologize for having missed this work on Sobolev training in discussing related work. Following the excellent suggestions of the reviewer, we have investigated Sobolev training in some detail and would like to explain our point of view on it below,
>
>     - First, in our reading of references [1-4] mentioned by you, the Sobolev training refers to using $H^r$ norms with $r \geq 1$ for the PDE residual, rather than the $L^2$ norm, as the physics-informed loss function. These papers have also shown that using this loss can improve training. However, we were unable to see how exactly these papers consider *preconditioning* in a precise sense as it is unclear to us what exactly was the underlying condition number. Thus, there is a clear difference in our approach where we precisely define the condition number of the underlying operator that composes the Hermitian square of the differential operator with the Tangent Kernel operator.
>     - It turns out that we can also analyze Sobolev training within the operator preconditioning framework that we have formulated in our paper. To this end, we revisit the example considered in Theorem 3.1 of our paper concerning the Laplacian in one space dimension as calculations can be explicitly performed in this case. We simply repeat the steps of the proof to account for the $H^r$-Sobolev norm. The matrix $\mathbb{A}$ is now given as $\mathbb{A}^* = \mathbb{D}^* + \lambda vv^\top$ with $\mathbb{D}^* := \sum_{\ell=0}^r \langle \mathcal{D} \partial_x^\ell \phi, \mathcal{D} \partial_x^\ell \phi\rangle_{L^2}$. One can then calculate that $\mathbb{D}^*$ is diagonal with $\mathbb{D}^*_{kk} = \sum_{\ell=0}^r k^{4+2\ell}$. An analogous calculation to the one in Theorem 3.1 then shows that the lower bound is $\kappa(\mathbb{A}^*) \geq K^4 \cdot \tfrac{1}{r+1}\sum_{\ell=0}^r K^{2\ell}$, which is clearly a worse lower bound than without Sobolev training, i.e., with $r=0$. For instance, for $H^1$ we find that $\kappa(\mathbb{A}^*) \geq K^6/2$ and for $H^2$ we find that  $\kappa(\mathbb{A}^*) \geq K^8/3$, which is considerably worse.
>     - We also ran numerical experiments, for both Fourier features as well as MLPs using Sobolev training. The comparative results with/without Sobolev training are provided in the Table below:
>
>         | Model  | Epochs  | Total loss  | Relative $L^1$ error | Condition number  |
>         |---|---|---|---|---|
>         | ff_no_precond_no_sobolev  | 2000  | 0.01537   | 0.02386   | 31520.74    |
>         | ff_no_precond_sobolev | 2000 | 3750.37 | 0.15199 | 1536899  |
>         |  ff_precond_sobolev |  20 | $\approx 3.37 \times 10^{-32}$ | $\approx 5.29 \times 10^{-16}$|  1.16   |
>         | mlp_no_precond_no_sobolev | 2000 |  0.05171 |  0.02653 | - |
>         | mlp_no_precond_no_sobolev | 2000  |  5360540 | 1.29184 | - |
>
>         Here, Sobolev training uses the $H^1$-norm. As we see from the table, for both Fourier features and MLPs, Sobolev training significantly worsens the training landscape as well as the resulting test errors and the results are consistent with the theory elaborated above. Hence, even for this simple problem, Sobolev training does not seem to pay off nor can it be viewed as a *preconditioning* technique, when preconditioning is defined as in our framework.

---

> ### Author Response · Authors · 2023-11-18
> **Response to Reviewer E6qc (part 1)**
>
> We thank the reviewer for their detailed feedback on our paper. Below, we address the questions raised by the reviewer and thank the reviewer in advance for their patience in reading our reply.
>
> 1. (W1): The reviewer has pointed to **Missing related work**. We would like to thank the reviewer for identifying very interesting related work, especially on Sobolev training and competitive PINNs that we will certainly include in a **CRV**, if accepted. Although we have taken care to mention previous work on preconditioning in the field of numerical analysis, we apologize having missed the stated work in ML and will include them in a **CRV**, if accepted.
> 2. (W2). The reviewer has addressed the issue of **Sobolev Training** and its relation to preconditioning at great length. We apologize for having missed this work on Sobolev training in discussing related work. Following the excellent suggestions of the reviewer, we have investigated Sobolev training in some detail and would like to explain our point of view on it below,
>
>     - First, in our reading of references [1-4] mentioned by you, the Sobolev training refers to using $H^r$ norms with $r \geq 1$ for the PDE residual, rather than the $L^2$ norm, as the physics-informed loss function. These papers have also shown that using this loss can improve training. However, we were unable to see how exactly these papers consider *preconditioning* in a precise sense as it is unclear to us what exactly was the underlying condition number. Thus, there is a clear difference in our approach where we precisely define the condition number of the underlying operator that composes the Hermitian square of the differential operator with the Tangent Kernel operator.
>     - It turns out that we can also analyze Sobolev training within the operator preconditioning framework that we have formulated in our paper. To this end, we revisit the example considered in Theorem 3.1 of our paper concerning the Laplacian in one space dimension as calculations can be explicitly performed in this case. We simply repeat the steps of the proof to account for the $H^r$-Sobolev norm. The matrix $\mathbb{A}$ is now given as $\mathbb{A}^* = \mathbb{D}^* + \lambda vv^\top$ with $\mathbb{D}^* := \sum_{\ell=0}^r \langle \mathcal{D} \partial_x^\ell \phi, \mathcal{D} \partial_x^\ell \phi\rangle_{L^2}$. One can then calculate that $\mathbb{D}^*$ is diagonal with $\mathbb{D}^*_{kk} = \sum_{\ell=0}^r k^{4+2\ell}$. An analogous calculation to the one in Theorem 3.1 then shows that the lower bound is $\kappa(\mathbb{A}^*) \geq K^4 \cdot \tfrac{1}{r+1}\sum_{\ell=0}^r K^{2\ell}$, which is clearly a worse lower bound than without Sobolev training, i.e., with $r=0$. For instance, for $H^1$ we find that $\kappa(\mathbb{A}^*) \geq K^6/2$ and for $H^2$ we find that  $\kappa(\mathbb{A}^*) \geq K^8/3$, which is considerably worse.
>     - We also ran numerical experiments, for both Fourier features as well as MLPs using Sobolev training. The comparative results with/without Sobolev training are provided in the Table below:
>
>         | Model  | Epochs  | Total loss  | Relative $L^1$ error | Condition number  |
>         |---|---|---|---|---|
>         | ff_no_precond_no_sobolev  | 2000  | 0.01537   | 0.02386   | 31520.74    |
>         | ff_no_precond_sobolev | 2000 | 3750.37 | 0.15199 | 1536899  |
>         |  ff_precond_sobolev |  20 | $\approx 3.37 \times 10^{-32}$ | $\approx 5.29 \times 10^{-16}$|  1.16   |
>         | mlp_no_precond_no_sobolev | 2000 |  0.05171 |  0.02653 | - |
>         | mlp_no_precond_no_sobolev | 2000  |  5360540 | 1.29184 | - |
>
>         Here, Sobolev training uses the $H^1$-norm. As we see from the table, for both Fourier features and MLPs, Sobolev training significantly worsens the training landscape as well as the resulting test errors and the results are consistent with the theory elaborated above. Hence, even for this simple problem, Sobolev training does not seem to pay off nor can it be viewed as a *preconditioning* technique, when preconditioning is defined as in our framework.

---

> ### Author Response · Authors · 2023-11-18
> **Response to Reviewer E6qc (part 2)**
>
> 2. (continued)
>     - Next, the papers like [5, 8-10] also (essentially) suggest to use an $H^r$-norm during training but now with *$r<0$*, in contrast with [1-4]. We agree with the reviewer that this could be a valid method of preconditioning. Again, we analyze it from the preconditioning perspective suggested by us. Repeating the analysis for the Poisson equation above by applying $(I-\Delta)^{-1}$ to the operator $\mathcal{D}$, the matrix $\mathbb{A}$ is now given as $\mathbb{A}^* = \mathbb{D}^* + \lambda vv^\top$ with $\mathbb{D}^* := \langle (I-\Delta)^{-1}\mathcal{D} \phi, (I-\Delta)^{-1}\mathcal{D}  \phi\rangle_{L^2}$. One can then calculate that $\mathbb{D}^*$ is diagonal with $\mathbb{D}^*_{kk} = k^4/(1+k^2)^2$. An analogous calculation to the one in Theorem 3.1 then shows that the lower bound is $\kappa(\mathbb{A}^*) \geq 4K^4/(1+K^2)^2$. This is a major improvement over the Fourier model without any preconditioning, but it can not achieve condition numbers close to 1 (like in Theorem 3.1) as for large $K$ the lower bound will be approximately 4.
>
>         Nevertheless, we would like to point that in practice, computing Sobolev norms with negative orders (or computing $(I-\Delta)^{-1}(\mathcal{D} u_\theta-f))$ can be very hard, even of comparable difficulty as solving the PDE. Hence, it is unclear to us if Sobolev training with negative Sobolev norms is going to be useful in solving PDEs.
>
>     Given that above discussion, we would like to emphasize that while Sobolev training might be a viable strategy in some contexts, we cannot support the reviewer's contention that **Sobolev training is equivalent to preconditioning**, let alone a preconditioning strategy that works in general. To function effectively as a preconditioner, it is crucial that the preconditioning approach adapts when dealing with different PDEs, as detailed in the paper. For instance, the operator effective in preconditioning the Poisson equation is unlikely to be suitable for the advection equation. This aspect is key and as Sobolev regularization is fixed it may not behave well for any given PDE.
>
>     Hence, we suggest that we will certainly mention Sobolev training and include the suggested references as another example of a preconditioning strategy in a **CRV**, if accepted. If the reviewer so wishes, we are happy to include an elaborate discussion on Sobolev training with the above results in the **CRV**. We thank the reviewer again for pointing out this avenue which we believe further strengthens the contributions of our paper.
> 3. Regarding the reviewer's question about **our contributions**,  we would like to explicitly point out the context and contributions of the paper. To this end, we take the liberty to repeat the relevant portion of our reply to all the reviewers above here: At the outset, it is amply clear in the literature that a major issue with physics-informed machine learning models such as PINNs is the difficulty of training them. This has been widely recognised and several ad-hoc strategies have been proposed to improve training. However, it is also clear that a suitable theoretical framework that precisely formulates the problem of optimizing physics-informed loss functions was missing. We believe that a lack of precision in defining what the problem actually is also impedes finding suitable measures for ameliorating the problem itself.
>
>     With this background in mind, our main aim in the paper was precisely to propose a suitable theoretical framework in which the issues affecting training of physics-informed machine learning models can be formulated and analyzed. To this end, we have rigorously connected the rate of convergence of gradient descent algorithms for physics-informed loss functions to the condition number of an operator, which involves the Hermitian-square of the underlying differential operator. This contribution is novel to the best of our knowledge and has been acknowledged as such by most of the reviewers. A corollary of this observation is the fact that preconditioning (now defined rigorously as reducing the condition number of the underlying operator) is necessary to accelerate training of physics-informed machine learning models. Again, this provides a mathematically precise definition of preconditioning for these models and stands in stark contrast to the existing literature where the words *preconditioning* and *condition numbers* are used very loosely and imprecisely. These two aforementioned aspects constitute our main contribution and has been emphasised as such repeatedly in our paper.

---

> ### Author Response · Authors · 2023-11-18
> **Response to Reviewer E6qc (part 3)**
>
> 3. (continued) Given the precise notion of preconditioning that we propose, it was incumbent upon us to also analyse several existing strategies for improving training in physics-informed machine learning through the perspective of our framework of operator preconditioning. To this end, we have in Section 3 and in the **SM**, analyzed via a combination of theoretical analysis and empirical evaluation, several such strategies (Choice of balance parameter $\lambda$, Hard Boundary conditions, Second-order optimizers, Domain decomposition and Casual learning, we will also include *Sobolev Training*) through the lens of operator preconditioning, providing a novel interpretation of these strategies and identifying conditions under which they might or might not work. We had gone further and analyzed an additional strategy that essentially boils down to *rescaling Fourier features*, whether with respect to linear basis functions or nonlinear neural networks. These are secondary contributions and we feel that it is not fair to judge the article based on secondary, rather than primary contributions.
>
>     In response to the reviewer's points about **lack of general preconditioning strategies**, we highlight Sec 3 where many existing strategies are examined through the lens of operator preconditioning. Moreover, in **SM* Section C.3, we have additionally explored another general simple preconditioning strategy which consists of using an approximate inverse of the matrix $\mathbb{A}$ (Eqn. (2.6)) as a preconditioner for the gradient descent step, i.e., pre-multiplying $(\mathbb{A} + \varepsilon I)^{-1}$ to the gradient descent update. This strategy is also applicable to any PDE and any linear or nonlinear model.
> 4. (W3) We thank the reviewer for their careful inspection of our paper and have edited all typos pointed out by you. In addition, we will expand the bibliography following their suggestions in the **CRV**, if accepted.
> 5. (Q1) We have discussed *the main difference between the preconditioning method proposed in this paper and sobolev training* and *Is there any numerical example that your method outperforms (higher accuracy, lower computational cost, etc) sobolev training?* in some detail above.
>
> We sincerely hope that we have addressed all the reviewer's concerns, particularly about the relation between our paper and the literature on Sobolev training, to your satisfaction. We kindly request the reviewer to upgrade your assessment of our paper accordingly.

---

> > ### Comment · Reviewer_E6qc · 2023-11-20
> > **Response to the authors' rebuttal**
> >
> > The reviewer would like to thank the authors for their detailed response, which has addressed the reviewer's main question about the relation between Sobolev training and preconditioning. Just one more comment: the authors are encouraged to have a subsection discussing previous work on preconditioning from both the ML community (such as [7] above) and the applied math community (such as this classical paper: Rheinboldt, W. C. (1976). On measures of ill-conditioning for nonlinear equations. Mathematics of Computation, 30(133), 104-111).
> >
> > The reviewer agrees that the general framework of using preconditioning to accelerate PINN training proposed by the authors is valuable and will be beneficial for the scientific machine learning community. Therefore, I have increased my score from 5 to 6. Please let me know if you have any further question.

---

> > > ### Author Response · Authors · 2023-11-20
> > > **Thanking the reviewer**
> > >
> > > We thank the reviewer for your prompt feedback and for increasing your rating of our paper. Your point about discussing preconditioning in classical numerical analysis through the Rheinbolt paper is well-taken and we would consider do so in the **CRV**, if accepted.

---

### Official Review · Reviewer_95TT · 2023-10-30

**Soundness:** 2 fair
**Presentation:** 3 good
**Contribution:** 2 fair
**Rating:** 5
**Confidence:** 3

**Summary:**

This paper considers the problem of training neural networks to solve partial differential equations (PDEs) using a physics-informed neural networks (PINNs) approach. In the introduction, the authors mention empirical studies suggesting that training might be the bottleneck for achieving robust convergence in PINNs, which motivates their approach of studying the behavior of gradient descent algorithms in a PINNs context. Then, in section 2, the authors study a simplification of a gradient step and show that the rate of convergence depends on the condition number of a Gram matrix associated with the PDE. This motivates the authors approach for preconditioning the operator to achieve better convergence rates in practical applications.

**Strengths:**

- The paper is well-written and focuses on an important problem in PINNs.
- The authors provide a clear exposition of their approach with both theoretical and empirical results.

**Weaknesses:**

- On p.3, the authors mention that their aim is to analyze the convergence of gradient descent. However, the analysis in section 2 only considers a simplified gradient step where they neglect all higher order terms in an error term epsilon_k, which they assume to be small. Is that equivalent to performing a local analysis close to a local minima and if so, how does the current analysis relate to the NTK analysis in the context of PINNs (Wang et al., 2022)? It is not clear whether this explains the poor performance of PINNs in practice as the main issue is to understand the pre-asymptotic behavior of gradient descent.
- Thm. 2.4 shows that the condition number of the Gram matrix is greater than the condition number of a Hermitian square operator A\circ TT^*. However, to perform preconditioning, one should instead derive an upper bound on the condition number of the Gram matrix. I am aware that Thm 2.4 gives an equality case but it assumes that the Gram matrix is invertible.
- The practical section 3 is the main weakness of the paper. It seems that the authors approach might only be useful when one already know a really good approximant to the eigenvectors and eigenvalues of the PDE, when it is linear (e.g. the Poisson equation example in Fig. 1). In this case, why would one use a neural network to solve the PDE instead of a spectral method? The authors states that they employ a linear combination of eigenvectors of the Poisson equation as the machine learning model, but this is exactly a spectral method and we know that the performance of an iterative solver (e.g. conjugate gradient, GMRES) for solving the system is related to the condition number of the matrix. The authors give very little details on how one might use their approach in a real context with a complex neural networks to solve a nonlinear PDE.

**Questions:**

- In eq. (2.6), the function space should be defined so that D\phi_i lies in L^2.
- I would suggest using \bm notation for matrices and vectors instead of \mathbb, which could be confused with the set of complex numbers for \mathbb{C}.
- Lemma 2.2, is there a way of controlling the error epsilon_k with respect to m for large m?
- Typo: Lemma 2.1: "Let If eta = ..."
- Typo: p.7 "precondiioned Fourier model"
- Fig. 3 (left): I don't really see an improvement of the approach as their are still a large number of 0 eigenvalues with the MLP + preconditioned Fourier features approach, suggesting that the system is ill-conditioned.
- The related work section would be better placed in the introduction.

---

> ### Author Response · Authors · 2023-11-18
> **Reply to Reviewer 95TT (part 1)**
>
> We start by thanking the reviewer for their appreciation of the strengths of our theoretical and empirical strengths of our paper. Below, we address the questions raised by the reviewer and thank the reviewer in advance for their patience in reading our detailed reply.
>
> 1. (W1) Regarding the reviewer's point that **they neglect all higher order terms in an error term $\epsilon_k$, which they assume to be small**, we want to highlight that we do not just assume that this error term is small, but that rather we show that it is identically zero for linear models and that we prove rigorously in Lemma 2.2 that this error term is small in the NTK regime for neural networks. Furthermore, the reviewer asks **Is that equivalent to performing a local analysis close to a local minima**: we stress that this is not the setting of our analysis, but rather our analysis requires the smallness of the error term in Lemma 2.1. This error term is exactly zero for linear models and can be made small (Lemma 2.2) in the NTK regime for neural networks. There are no further assumptions. Hence, our analysis also covers the pre-asymptotic behavior of gradient descent as we explicitly consider the discrete time setting with finitely many parameters. This is precisely the setting where condition numbers make sense.
> 2. (W2) Regarding the reviewer's question on **However, to perform preconditioning, one should instead derive an upper bound on the condition number of the Gram matrix**, we fully agree with the reviewer that this general upper bound does not prove that a well-conditioned matrix implies a well-conditioned operator $\mathcal{A}\circ TT^*$. Fortunately, the equality holds for many practical cases. For linear basis functions the Gram matrix is trivially invertible, and also for randomly initialized neural networks (e.g. using normal distribution) it will be invertible as the $\phi_i$ will be independent functions with probability 1, and hence equality will also hold. Finally, in practice it is the conditioning of the matrix $\mathbb{A}$ that determines the convergence of gradient descent, and not directly that of the operator $\mathcal{A}\circ TT^*$, so the condition number of the matrix $\mathbb{A}$ is much more important in preconditioning in practice.
> 3. (W3) Regarding the reviewer's comment that **The practical section 3 is the main weakness of the paper**, we would like to explicitly point out the context and contributions of the paper. To this end, we take the liberty to repeat the relevant portion of our reply to all the reviewers above here: At the outset, it is amply clear in the literature that a major issue with physics-informed machine learning models such as PINNs is the difficulty of training them. This has been widely recognised and several ad-hoc strategies have been proposed to improve training. However, it is also clear that a suitable theoretical framework that precisely formulates the problem of optimizing physics-informed loss functions was missing. We believe that a lack of precision in defining what the problem actually is also impedes finding suitable measures for ameliorating the problem itself.
>
>    With this background in mind, our main aim in the paper was precisely to propose a suitable theoretical framework in which the issues affecting training of physics-informed machine learning models can be formulated and analyzed. To this end, we have rigorously connected the rate of convergence of gradient descent algorithms for physics-informed loss functions to the condition number of an operator, which involves the Hermitian-square of the underlying differential operator. This contribution is novel to the best of our knowledge and has been acknowledged as such by most of the reviewers. A corollary of this observation is the fact that preconditioning (now defined rigorously as reducing the condition number of the underlying operator) is necessary to accelerate training of physics-informed machine learning models. Again, this provides a mathematically precise definition of preconditioning for these models and stands in stark contrast to the existing literature where the words *preconditioning* and *condition numbers* are used very loosely and imprecisely. These two aforementioned aspects constitute our main contribution and has been emphasised as such repeatedly in our paper.

---

> ### Author Response · Authors · 2023-11-18
> **Reply to Reviewer 95TT (part 2)**
>
> 2. (continued) Given the precise notion of preconditioning that we propose, it was incumbent upon us to also analyse several existing strategies for improving training in physics-informed machine learning through the perspective of our framework of operator preconditioning. To this end, we have in Section 3 and in the **SM**, analyzed via a combination of theoretical analysis and empirical evaluation, several such strategies (Choice of balance parameter $\lambda$, Hard Boundary conditions, Second-order optimizers, Domain decomposition and Casual learning) through the lens of operator preconditioning, providing a novel interpretation of these strategies and identifying conditions under which they might or might not work. We had gone further and analyzed an additional strategy that essentially boils down to *rescaling Fourier features*, whether with respect to linear basis functions or nonlinear neural networks. These are secondary contributions and we feel that it is not fair to judge the article based on secondary, rather than primary contributions.
>
>    Regarding the reviewer's specific point about **Spectral methods for the Poisson Equation**, we would like to point that preconditioning for spectral methods is indeed well-established. However, there are essential differences between our approach and using a spectral method. Because, we minimize the loss in an $L^2$-sense, we are confronted with having to analyze the spectrum of the *Hermitian-square* of the underlying differential operator, where as in a conventional spectral method, only the differential operator itself has to be pre-conditioned. Following the reviewer's excellent suggestion, we will explicitly mention the differences with a spectral method in a **CRV**, if accepted.
>
>    In response to the reviewer's points about **lack of general preconditioning strategies**, we highlight Sec 3 where many existing strategies are examined through the lens of operator preconditioning. Moreover, in **SM* Section C.3, we have additionally explored another general simple preconditioning strategy which consists of using an approximate inverse of the matrix $\mathbb{A}$ (Eqn. (2.6)) as a preconditioner for the gradient descent step, i.e., pre-multiplying $(\mathbb{A} + \varepsilon I)^{-1}$ to the gradient descent update. This strategy is also applicable to any PDE and any linear or nonlinear model.
> 4. (Q3) Regarding the reviewer's question on **Lemma 2.2, is there a way of controlling the error $\epsilon_k$ with respect to $m$ for large $m$?** Our analysis for particular Lemma builds on the results from Wang et. al, which do not control the error in terms of $m$. However, a careful examination of the proof reveals that this precise control is likely to be possible, yet very tedious. We will attempt to perform this analysis and include it in a **CRV**, if accepted and thank the reviewer for pointing it to us.
> 5. (Q6) In response to the reviewer's question on **Fig. 3 (left): I don't really see an improvement of the approach as there are still a large number of 0 eigenvalues with the MLP + preconditioned Fourier features approach, suggesting that the system is ill-conditioned**, we would like to point out that he spectrum of the preconditioned model is much more localized, which is often correlated to better convergence as seen in Figure 3 (right).
> 6. We have addressed all the other minor concerns of the reviewer in the modified version of the paper.
>
> We sincerely hope that we have addressed all the reviewer's concerns to your satisfaction. We kindly request the reviewer to upgrade your assessment of our paper accordingly.

---

> > ### Author Response · Authors · 2023-11-21
> > **Requesting the reviewer for feedback**
> >
> > Due to the imminent closure of the discussion period, We kindly request the reviewer to provide us some feedback on our rebuttal. We are at your disposal for any further questions/clarifications regarding the paper or the rebuttal.

---

> > > ### Comment · Reviewer_95TT · 2023-11-21
> > >
> > > I thank the authors for the detailed response which addressed most of my comments.
> > >
> > > I am however not convinced by the answer to Q.6: the whole point of the preconditionning approach is to improve the condition number of A. However, looking at Fig. 3, it seems to me that the condition number remains the same between the MLP and MLP+preconditioned Fourier features (same lowest and largest eigenvalues). I would have expected a similar transformation of the spectrum like the blue eigenvalues to explain the difference in training on the right plot. As it currently stands, I do not think that Fig. 3 really demonstrates that the preconditioning approach is an explanation for the fast loss decay for nonlinear models.

---

> ### Author Response · Authors · 2023-11-22
> **Reply to the Reviewer**
>
> At the outset, we would like to sincerely thank the reviewer for reading our response and providing constructive feedback on it. We are grateful to hear that most of the reviewer's concerns are addressed and we thank them for the clarification of the final question. Regarding Q6, we respond below,
>
> First, we would like to make an important point regarding conditioning of the matrix A, the condition number, and its relationship to convergence of gradient descent. The reviewer is absolutely correct that--as a consequence of having near zero eigenvalues in both the MLP and MLP + preconditioned Fourier features case (as shown in Fig. 3) , the minimum and maximum eigenvalue are approximately the same, hence they yield approximately the same condition number. Although a smaller condition number is a *sufficient* condition for faster convergence, it is not a *necessary* condition, at least for the pre-asymptotic convergence. Indeed,  if we decompose the difference $\widetilde{\theta_0}-\vartheta = \sum_i \alpha_i v_i$ along the eigenspaces of $\mathbb{A}$, we can rewrite equation **SM** (A.16) as follows:
> $$\widetilde{\theta_k}-\vartheta = (I-\eta \mathbb{A})^k(\widetilde{\theta_0}-\vartheta) = \sum_{i=1}^n (1-\eta\lambda_i)^k \alpha_i v_i,$$
> where $\lambda_{\min} := \lambda_1 \leq \lambda_2 \leq \ldots \leq \lambda_n =: \lambda_{\max}$ the $n$ eigenvalues of $\mathbb{A}$ and $v_1, \ldots, v_n$ the corresponding eigenvectors. This is  a refinement of Theorem 2.3. Hence by recalling that $\eta = c/\lambda_{\max}$ and developing the argument of **SM** Eqn (A.17) and (A.18), we observe that the more eigenvalues are closer to $\lambda_{\max}$, the more terms in the above sum are closer to 0 and the smaller $\Vert \widetilde{\theta_k}-\vartheta\Vert_2$ will be. In Figure 3 this is the case going from the MLP to the MLP + Preconditioned Fourier features, and even more so when considering the preconditioned Fourier features. We believe this finer grained perspective is very useful when thinking about convergence, and thus we will take care to discuss in the **CRV** of our paper, if it is accepted.
>
>  In addition, regarding the comment **I would have expected a similar transformation of the spectrum like the blue eigenvalues to explain the difference in training on the right plot**, we point to the fact that we only presented the spectrum resulting from preconditioning the Fourier features within the MLP in this particular figure and did not precondition the MLP as a whole. As a result, the improvement in the spectrum is not as large, and also not expected to be as large, as for the Fourier features without MLP, which is in complete correspondence with the observation that the preconditioned Fourier features (without MLP) achieve much lower errors and converge much faster than the MLP with preconditioned Fourier features (see Figure 1).
>
> However, with your feedback, along with feedback from other reviewers we were able to push preconditioning  further in the revised version, and have corroborated our findings in the fully non-linear setting by preconditioning by  pre-multiplying $(\mathbb{A} + \varepsilon I)^{-1}$ to the gradient descent update (see Section C.3 of **SM**). We have now added an additional Figure 18 to the **SM** where we plot the spread of the eigenvalues and convergence of the loss, in the MLP and preconditioned-MLP case. In this case, the result is much clearer: although there is still quite a few (near) zero eigenvalues (the zero eigenvalues originate from the fact that the PINNs loss with an MLP is highly ill-conditioned), there are much more eigenvalues near $\lambda_{\max}$  for the preconditioned-MLP, and as a consequence of this and following the above argument, training is indeed much faster.
>
> Needless to say, we will add these results and the corresponding discussion to the **CRV**, if accepted and thank the reviewer again for pointing out this avenue of improvement to us. We hope to have addressed your remaining concern satisfactorily and if so, we would kindly request the reviewer to update their assessment accordingly.

---

### Official Review · Reviewer_2Mn2 · 2023-10-31

**Soundness:** 3 good
**Presentation:** 3 good
**Contribution:** 3 good
**Rating:** 6
**Confidence:** 3

**Summary:**

The paper investigates behaviors of gradient descent algorithms when the loss term involves minimizations of the residual connected to partial differential equations (PDEs). The paper simplifies the gradient descent rules via utilizing the Taylor expansion and derives the relationship between the rate of convergence and the differential operator. In particular, the paper provides theoretical analysis showing that the rate of convergence depends on the conditioning of an operator denoted by ``Hermitian square’’, which is derived from the differential operator and its adjoint. The paper proposes a method for preconditioning the differential operator and tests the algorithms on benchmark problems.

**Strengths:**

- The paper is well-written and theoretically sound (although there are some typos, most notably, there is no $\lambda$ in Eq. 2.2).

- The paper tackles an important issue in minimizing PDE residual losses, which is not popularized in many scientific machine learning applications.

**Weaknesses:**

- It is less convincing if the proposed method can be applicable to a wide variety of neural networks, which is also pointed out in the manuscript, “preconditioning A for nonlinear models such as neural networks can, in general, be hard” on page 7 and in the limitation paragraph. In the manuscript, preconditioning strategies are showcased only with the simple cases, approximating solutions of Poisson equations via linear and nonlinear models with Fourier feature mapping. It does not seem that the paper provides some guidelines or insights for preconditioning strategies for general cases.

- Computational aspects are weak. In particular, comparisons with other works improving optimizers of PINNs in more general classes of PDEs, which also include the assessment on computational wall times, would be more appreciated and informative for readers/users to decide which method to use. Even if it’s not computational comparisons, it would be better to have some descriptions on differences from other baselines [2,3] (as shown in page 8, where the connections to [1] is presented in the Choice of lambda paragraph.)

[1] Wang, et al, 2022, JCP (the reference in the paper)

[2] Basir and Senocak, Physics and equality constrained artificial neural networks: Application to forward and inverse problems with multi-fidelity data fusion, 2022, JCP

[3] Kim et al, DPM: A novel training method for physics-informed neural networks in extrapolation, 2021, AAAI.

- Although the problem that is investigated in the paper is an important problem and the authors provide good contributions in the theoretical side, it is less convincing that the paper makes significant contributions in ML or DL perspectives.

**Questions:**

- could the authors provide more information/insight on how to build preconditioner for more general cases? for complex PDEs where the PINN does not employ Fourier feature transform?

- In the implementation, the gradient descent rule has been reimplemented following Eq. (3.2)?

---

> ### Author Response · Authors · 2023-11-18
> **Reply to Reviewer 2Mn2**
>
> We thank the reviewer for acknowledging the importance of the problem our paper addresses, and for their other positive comments. Below, we address the questions raised by the reviewer and thank the reviewer in advance for their patience in reading our detailed reply.\\\\
> 1. Regarding the reviewer's comment about us explicitly mentioning **preconditioning A for nonlinear models such as neural networks can, in general, be hard**, we start by clarifying what me meant in this context, namely that, in the neural network setting the operator $TT^*$ appearing in the main theorem 2.4 does not reduce to a simple operator like it does in the case of Fourier features or other linear basis functions. This implies that the condition number of the operator $\mathcal{A} \circ TT^*$ is  more difficult to study, but by no means is it impossible
> as we consider in Section 3 as well as in **SM** Section C.
> 2. Regarding the reviewer's concern that **It does not seem that the paper provides some guidelines or insights for preconditioning strategies for general cases** and the question **how to build preconditioner for more general cases**, we start by repeating our detailed reply to all the reviewers providing the context of our contributions.  At the outset, it is amply clear in the literature that a major issue with physics-informed machine learning models such as PINNs is the difficulty of training them. This has been widely recognised and several ad-hoc strategies have been proposed to improve training. However, it is also clear that a suitable theoretical framework that precisely formulates the problem of optimizing physics-informed loss functions was missing. We believe that a lack of precision in defining what the problem actually is also impedes finding suitable measures for ameliorating the problem itself.
>
>    With this background in mind, our main aim in the paper was precisely to propose a suitable theoretical framework in which the issues affecting training of physics-informed machine learning models can be formulated and analyzed. To this end, we have rigorously connected the rate of convergence of gradient descent algorithms for physics-informed loss functions to the condition number of an operator, which involves the Hermitian-square of the underlying differential operator. This contribution is novel to the best of our knowledge and has been acknowledged as such by most of the reviewers. A corollary of this observation is the fact that preconditioning (now defined rigorously as reducing the condition number of the underlying operator) is necessary to accelerate training of physics-informed machine learning models. Again, this provides a mathematically precise definition of preconditioning for these models and stands in stark contrast to the existing literature where the words *preconditioning* and *condition numbers* are used very loosely and imprecisely. These two aforementioned aspects constitute our main contribution and has been emphasised as such repeatedly in our paper.
>
>    Given the precise notion of preconditioning that we propose, it was incumbent upon us to also analyse several existing strategies for improving training in physics-informed machine learning through the perspective of our framework of operator preconditioning. To this end, we have in Section 3 and in the **SM**, analyzed via a combination of theoretical analysis and empirical evaluation, several such strategies (Choice of balance parameter $\lambda$, Hard Boundary conditions, Second-order optimizers, Domain decomposition and Causual learning) through the lens of operator preconditioning, providing a novel interpretation of these strategies and identifying conditions under which they might or might not work. We had gone further and analyzed an additional strategy that essentially boils down to *rescaling Fourier features*, whether with respect to linear basis functions or nonlinear neural networks. These are secondary contributions and we feel that it is not fair to judge the article based on secondary, rather than primary contributions.

---

> ### Author Response · Authors · 2023-11-18
> **Reply to Reviewer 2Mn2 (part 2)**
>
> 2. (continued) Moreover, the fact that only linear PDEs was considered in the empirical evaluation was based on the observation that if physics-informed machine learning models do not work for simple linear PDEs, there is little hope of them working for complicated nonlinear PDEs. Furthermore, we follow widely cited papers such as Krishnapriyan et al, 2021, which flagged the problems of training PINNs by focussing considerably on simple linear PDEs. Finally, the page limit also automatically limits the scope of an article. Given these considerations, we believe that linear PDEs already provide a reasonably representative arena for analyzing the issue of training PINNs and related models and lessons learnt there can be applied to nonlinear PDEs in future works. In particular, at some level, nonlinear PDEs have to be **linearized** and the lessons learnt from linear PDEs will be germane. Finally in this context, we would like to mention that there is a long tradition of analyzing training in machine learning through linear models (see ArXiv:1810.02281 and references therein) as analyzing gradient descent even with linear models is highly non-trivial and representative of the difficulties faced in training non-linear models.
>
>    Finally, we have added a new section **SM** Section C.3, where we consider an additional and very general simple preconditioning strategy which consists of using an approximate inverse of the matrix $\mathbb{A}$ (Eqn. (2.6)) as a preconditioner for the gradient descent step, i.e., pre-multiplying $(\mathbb{A} + \varepsilon I)^{-1}$ to the gradient descent update.
> 3. Following the reviewer's excellent suggestion, we have now included **wall-clock times for the methods in **SM** Table 2. The reviewer's suggestion to discuss Refs. [2] and [3] are well-taken and would be included in a **CRV**, if accepted. In this context, we would like to comment that the contributions of these papers can be viewed through a preconditioning perspective as done for the contributions of your reference [1] as described in **SM** section B.2.
> 4. Regarding the reviewer's concern about **it is less convincing that the paper makes significant contributions in ML or DL perspectives**, please refer to our detailed framing of the contributions as we have outlined them to you in point 2. above.
> 5. Regarding the reviewer's question on **how the gradient descent rule has been reimplemented following Eq. (3.2)**, we would like to point out that we have shown that rescaling the model parameters is equivalent to preconditioning the gradient directly. Hence, in our experiments with linear models or MLP with Fourier features, we rescaled our model parameters and therefore did not use (3.2) directly. On the other hand, with the matrix inversion strategy of **SM** Sec C.3, we use a form of Eqn (3.2) for the gradient update.
>
> We sincerely hope that we have addressed all the reviewer's concerns, particularly about the framing of our contributions, to your satisfaction. We kindly request the reviewer to upgrade your assessment of our paper accordingly.

---

> > ### Author Response · Authors · 2023-11-21
> > **Requesting the reviewer for feedback**
> >
> > Due to the imminent closure of the discussion period, We kindly request the reviewer to provide us some feedback on our rebuttal. We are at your disposal for any further questions/clarifications regarding the paper or the rebuttal.

---

> > > ### Comment · Reviewer_2Mn2 · 2023-11-22
> > >
> > > I'd like to thank the authors for the response for the clarification and additional materials. While the reported wall time again raises some concerns on practicality, the theoretical contribution that the paper provides seems to outweigh those weaknesses.  Also, it feels to me that this paper opens up an important direction which needs further studies in the community. Lastly, considering that developing a single preconditioner for nonlinear systems of equations arising from specific equations e.g., Stokes, NS problems, can be considered as a significant contributions, requesting some guidelines for building preconditioners that work generally well for all cases could be too much to ask. Given these points, I update the rating accordingly.

---

> > > > ### Author Response · Authors · 2023-11-22
> > > > **Thanking the Reviewer**
> > > >
> > > > We sincerely thank the reviewer for your their feedback as well as for raising our score. Your comments were very helpful in improving the quality of our contributions.

---

### Official Review · Reviewer_gVcB · 2023-10-31

**Soundness:** 4 excellent
**Presentation:** 3 good
**Contribution:** 4 excellent
**Rating:** 8
**Confidence:** 4

**Summary:**

This paper proposes a preconditioning technique for solving optimisation problems associated with solving ODE, which can be expressed through the differential operator used and some kernel integral operator which is variable and can be chosen to minimize the impact of conditioning number of the optimisation problem. This approach appears to be practically efficient for model of Fourier features.

**Strengths:**

The proposed preconditioning is elegantly inferred from the approximate form of iteration so that the approach seems to be natural. Theoretical framework allows obtaining improved convergence rates with no dependence on conditioning number of original problem, which is the issue even in finite-dimensional optimisation. Practical efficiency of the precodnitioning operation in its characterising property, decreasing conditioning number, was demonstrated.

**Weaknesses:**

No significant weakness that I can notice as a specialist in finite-dimensional optimisation. What was described is quite earnestly and helpful for future readers.

**Questions:**

I assume that there are models for which kernel integral operator cannot be chosen manually. What you propose to do if expressing precodnitioned operator explicitly is not possible? What do you think about applying additional optimisation procedure for training this kernel operator?

---

> ### Author Response · Authors · 2023-11-18
> **Reply to Reviewer gVcB**
>
> We thank the reviewer for their positive evaluation of our paper. Below, we address the questions raised by the reviewer and thank the reviewer in advance for their patience in reading our reply.
>
> 1. Regarding the reviewer's question about **What to do if the kernel integral operator cannot be chosen manually**: we agree that it is indeed often not possible to chose the kernel integral operator $TT^*$ explicitly. In this case, one can consider the various preconditioning techniques listed and analysed in the last paragraphs of Section 3 as well those in **SM**, Section C.3, which can always be applied.
> 2. The reviewer's sugggestion to **add another optimization layer to learn the kernel integral operator** is excellent. However, given the limited time for rebuttal, we will consider it as an interesting avenue for future research.
>
> We sincerely hope that we have addressed all the reviewer's concerns to your satisfaction.

---

### Official Review · Reviewer_Wdfd · 2023-11-01

**Soundness:** 4 excellent
**Presentation:** 3 good
**Contribution:** 4 excellent
**Rating:** 8
**Confidence:** 3

**Summary:**

The paper derives that under certain conditions, the gradient descent of the PINNs can be approximated using a simplified gradient descent algorithm, which further provides a linearized version of the training dynamics. The paper demonstrates that the speed of convergence of this simplified training dynamics depends on the condition number of an operator described using the underlying differential operator and the kernel integral operator. The paper provides a preconditioning approach that can almost achieve the ideal condition number of 1. The paper demonstrates the efficiency of preconditioning the operator approach for simple problems like linear Poisson and Advection Equation.

**Strengths:**

1. The paper derives a simplified gradient descent algorithm that linearizes the training dynamics, allowing easier analysis of the convergence speed.
2. The paper provides a strong theoretical foundation, and shows a novel result that the convergence of PINNs depends on the condition number of a certain operator.
3. Some of the results shown in the paper demonstrates super fast convergence (less than 50 epochs), which is quite impressive.

**Weaknesses:**

1. The paper assumes that the solution can be expressed as a linear combination of basis functions (such as fourier features, which was shown in the experiments). Such an approximation can be quite limiting, and the paper does not show how the preconditioning can be extended for non-linear models.
2. The evaluation of the proposed preconditioning approach is poor. Firstly, the paper only covers linear PDEs with linear PINN models. Second, only the training loss curves were shown for the two PDEs. It is well-known that the PINNs can learn trivial solutions and achieving very low training losses does not guarantee convergence. See paper [1].

[1] Daw, Arka, Jie Bu, Sifan Wang, Paris Perdikaris, and Anuj Karpatne. "Mitigating Propagation Failures in Physics-informed Neural Networks using Retain-Resample-Release (R3) Sampling." ICML 2023

**Questions:**

**Questions:**
1. For a typical Taylor series expansion, the Hessian would be computed at the point $\theta_0$. Can the authors provide some justification on why the Hessian was computed at an interpolated $\theta$ value between 0 and k?
2. Can the authors provide some discussions on the connections and differences of their approach and NTK theory, especially considering that the latter also offers a convergence rate analysis for PINNs?
3. For the Advection Equation with Fourier Features without preconditioning (Figure 2), it seems that the loss does not converge at all (the value of the loss is $10^3$). Can the authors provide some justification behind this? Is it because the model was chosen to be a simple linear one with Fourier Features?


**Minor Comments:**
1. Typo: Equation 2.2 the $\lambda$ is missing.
2. The Taylor expansion in Equation 2.5 should contain higher order terms or the paper should mention that the 3rd order and higher terms are ignored. Similarly, the proof in the appendix A.1 should mention that higher order multiplicative terms by substituting 2.5 in 2.2 are ignored.
3. The notation: $(\theta_k − \theta_0)^T H_k(\theta_k − \theta_0)$ can be confusing as it seems like the Hessian is computed at $H_k(\theta_k - \theta_0)$. The authors can consider using the following: $(\theta_k − \theta_0)^T H_k(x) (\theta_k − \theta_0)$

---

> ### Author Response · Authors · 2023-11-18
> **Reply to Reviewer Wdfd (part 1)**
>
> We start by thanking the reviewer for their appreciation of the strength of our theoretical foundation, and the very fast convergence when one uses this theory to properly precondition models, as you are among the few to highlight these contributions which we believe are of great value to the community. Below, we address the questions raised by the reviewer and thank the reviewer in advance for their patience in reading our detailed reply.
>
> 1. (W1) Regarding the point that **the paper assumes that the solution can be expressed as a linear combination of basis functions** and **the paper does not show how the preconditioning can be extended for non-linear models**: we start by pointing out that for our theoretical analysis, we only assume that the tangent kernel of the model is (approximately) constant along the optimization path, which can also hold for nonlinear models as neural networks. Infact, in Lemma 2.1, a precise charecterization of this error has been provided. Moreover, in the last paragraphs of Section 3 we have discussed and analysed many different  preconditioning approaches that also hold for nonlinear models. We have also added another simple preconditioning strategy which consists of using an approximate inverse of the matrix $\mathbb{A}$ (Eqn. (2.6)) as a preconditioner for the gradient descent step, i.e., pre-multiplying $(\mathbb{A} + \varepsilon I)^{-1}$ to the gradient descent update (see Section C.3 of **SM**) which is applicable to any nonlinear model.
> 2. (W2) Regarding the reviewer's concern that **the paper only covers linear PDEs with linear PINN models**,  we refer the reviewer to our discussion and experiments using neural networks in Section 3 as well as **SM** section C, and hence we do consider nonlinear models. Moreover, the fact that only linear PDEs was considered in the empirical evaluation was based on the observation that if physics-informed machine learning models do not work for simple linear PDEs, there is little hope of them working for general nonlinear PDEs. Furthermore, we burrowed from widely cited papers such as Krishnapriyan et al, 2021, which flagged the problems of training PINNs and focus considerably on simple linear PDEs. Finally, the page limit also automatically limits the scope of an article. Given these considerations, we believe that linear PDEs already provide a reasonably representative arena for analyzing the issue of training PINNs and related models and lessons learnt there can be applied to nonlinear PDEs in future works. In particular, at some level, nonlinear PDEs have to be **linearized** and the lessons learnt from linear PDEs will be germane. Finally in this context, we would like to mention that there is a long tradition of analyzing training in machine learning through linear models (see ArXiv:1810.02281) and references therein) as analyzing gradient descent even with linear models is highly non-trivial and representative of the difficulties faced in training non-linear models.
> 3. Regarding the reviewer's point that **only the training loss curves were shown for the two PDEs**, we follow the reviewer's excellent suggestion and plot the resulting $L^2$-test errors in the **SM** Figures 17-19, where the tight coupling between the residual losses and the test errors are demonstrated.
> 4. Regarding reviewer's (Q1+MC2) about **infinite Taylor series and ignoring higher-order terms**, we wish to clarify that we did not consider an infinite Taylor series and then ignored the higher-order terms. Instead, we considered a first-order Taylor polynomial with the corresponding *exact* remainder term (this is where the Hessian comes in and why it is evaluated at an interpolated value due to the mean-value theorem). We again stress that this formula is exact and no terms were ignored herein. We later show that the remainder term is small under suitable conditions (Lemma 2.2).
> 5. Regarding reviewer's (Q2) on the relation of our analysis with **NTK theory**, we would like to emphasize that NTK theory was originally formulated for the supervised MSE ($L^2$)-loss and for infinite-width neural networks. In contrast, we consider a physics-informed loss and finite-width neural networks, resulting in significant departures from the conventional NTK framework. Compared to other works that use NTK theory to analyse PINNs e.g. (Wang et al, 2022), we want to note that our analysis is valid in the practical case of *discrete time* rather than continuous time (as in Wang et. al.) and also we consider a finite number of parameters (corresponding to reality), along with the fact that we rigorously keep track of all error terms. Discrete time and finite parameters are absolutely crucial to our analysis as the condition number only appears in this discrete setting. Moreover, to the best of our knowledge, Wang et. al. do not identify an underlying operator and its conditioning as the key issues in training physics-informed machine learning models.

---

> ### Author Response · Authors · 2023-11-18
> **Reply to Reviewer Wdfd (part 2)**
>
> 6. Regarding the reviewer's Q3 about **non-convergence of unpreconditioned Fourier features for advection**, we start by pointing out that this cannot only be attributed to the *linear* form of the underlying model as the MLP also barely converges (see **SM** Figure 11). Rather, the ill-conditioning of the model is responsible and only we precondition the linear model, it converges very fast, as acknowledged by the reviewer. The non-convergence of MLPs in this case was also observed in Krishnapriyan et. al., 2021.
> 7. We have addressed the other minor comments of the reviewer in the revised version.
>
> We sincerely hope that we have addressed all the reviewer's concerns, particularly about use of linear vs. nonlinear models, to your satisfaction. We kindly request the reviewer to update your assessment of our paper accordingly.

---

> > ### Comment · Reviewer_Wdfd · 2023-11-20
> > **Response to Authors Rebuttal**
> >
> > I thank the authors for providing clarifications to my questions/comments. I have read the other reviews and the authors' comments, and I have increased my score accordingly.

---

> > > ### Author Response · Authors · 2023-11-21
> > > **Thanking the Reviewer**
> > >
> > > We sincerely thank the reviewer for their prompt feedback and for raising their score.

---

### Author Response · Authors · 2023-11-18
**Reply to All Reviewers**

At the outset, we would like to thank all the six reviewers for their thorough and patient reading of our article. Their fair criticism and constructive suggestions have enabled us to improve the quality of our article.

A revised version of the article and the **SM** is uploaded. We would also like to point out that all the references to page and line numbers, sections, figures, tables, equation numbers and references, refer to those in the revised version. Moreover, given that we have to synthesize 6 different opinions from reviewers, we would make further changes in a camera-ready version (**CRV**), if accepted, that will take into account the outcome of the discussion with all the reviewers.

We proceed to answer the points raised by each of the reviewers individually, below. Before that, we would like to address a point that was criticised by several of the reviewers. This point pertained to the fact that several reviewers mentioned that the **empirical results were weak** and **no general preconditioning strategy that can deal with nonlinear models for complex nonlinear PDEs was proposed**. We believe that this criticism might stem from a possible misunderstanding of the context and the contributions of our paper and would like to address it here. At the outset, it is amply clear in the literature that a major issue with physics-informed machine learning models such as PINNs is the difficulty of training them. This has been widely recognised and several ad-hoc strategies have been proposed to improve training. However, it is also clear that a suitable theoretical framework that precisely formulates the problem of optimizing physics-informed loss functions was missing. We believe that a lack of precision in defining what the problem actually is also impedes finding suitable measures for ameliorating the problem itself.

With this background in mind, our main aim in the paper was precisely to propose a suitable theoretical framework in which the issues affecting training of physics-informed machine learning models can be formulated and analyzed. To this end, we have rigorously connected the rate of convergence of gradient descent algorithms for physics-informed loss functions to the condition number of an operator, which involves the Hermitian-square of the underlying differential operator. This contribution is novel to the best of our knowledge and has been acknowledged as such by most of the reviewers. A corollary of this observation is the fact that preconditioning (now defined rigorously as reducing the condition number of the underlying operator) is necessary to accelerate training of physics-informed machine learning models. Again, this provides a mathematically precise definition of preconditioning for these models and stands in stark contrast to the existing literature where the words *preconditioning* and *condition numbers* are used very loosely and imprecisely. These two aforementioned aspects constitute our main contribution and has been emphasised as such repeatedly in our paper.

Given the precise notion of preconditioning that we propose, it was incumbent upon us to also analyse several existing strategies for improving training in physics-informed machine learning through the perspective of our framework of operator preconditioning. To this end, we have in Section 3 and in the **SM**, analyzed via a combination of theoretical analysis and empirical evaluation, several such strategies (Choice of balance parameter $\lambda$, Hard Boundary conditions, Second-order optimizers, Domain decomposition and Casual learning) through the lens of operator preconditioning, providing a novel interpretation of these strategies and identifying conditions under which they might or might not work. We had gone further and analyzed an additional strategy that essentially boils down to *rescaling Fourier features*, whether with respect to linear basis functions or nonlinear neural networks. These are secondary contributions and we feel that it is not fair to judge the article based on secondary, rather than primary contributions.

---

> ### Author Response · Authors · 2023-11-18
> **Reply to All Reviewers (part 2)**
>
> (see above for the first part )
>
> Moreover, the fact that only linear PDEs was considered in the empirical evaluation was based on the observation that if physics-informed machine learning models do not work for simple linear PDEs, there is little hope of them working for complicated nonlinear PDEs. Furthermore, we follow widely cited papers such as Krishnapriyan et al, 2021, which flagged the problems of training PINNs by focussing considerably on simple linear PDEs. Finally, the page limit also automatically limits the scope of an article. Given these considerations, we believe that linear PDEs already provide a reasonably representative arena for analyzing the issue of training PINNs and related models and lessons learnt there can be applied to nonlinear PDEs in future works. In particular, at some level, nonlinear PDEs have to be *linearized* and the lessons learnt from linear PDEs will be germane. Finally in this context, we would like to mention that there is a long tradition of analyzing training in machine learning through linear models (see ArXiv:1810.02281) and references therein) as analyzing gradient descent even with linear models is highly non-trivial and representative of the difficulties faced in training non-linear models.
>
> With these general comments, we hope to have addressed a common point of criticism of several of the reviewers and look forward to a healthy discussion.
>
> Yours sincerely,
>
> Authors of *An operator preconditioning perspective on training in physics-informed machine learning*

---

### Meta-Review · Area_Chair_ctSo · 2023-12-06

**Metareview:**

This paper offers a compelling new perspective on training in physics-informed machine learning, with a specific focus on the role of operator preconditioning. The authors propose that preconditioning a particular differential operator significantly impacts the efficiency of gradient descent algorithms used in these models' training. This claim is supported by robust theoretical analysis along with empirical evaluations. On the other hand, the paper has some limitations, particularly in its empirical parts. The evaluations primarily center around simple models, and there's a notable lack of extensive exploration into how the proposed preconditioning strategies could apply to more complex situations. This limitation somewhat narrows the immediate applicability and scope of the findings. Moreover, a more comprehensive comparison with existing methods could have provided clearer insights into the unique benefits and potential drawbacks of the proposed approach. During the discussion phase, the authors highlighted the theoretical nature of their contributions and their rationale for the initial focus on simple models. They also indicated plans to extend their methodology to more complex models in future work. While these responses addressed some of the concerns, skepticism remained about the practical implications of the findings.

In considering these aspects for the final decision, the theoretical contributions of the paper are significant. The insights provided about operator preconditioning in the context of training physics-informed models are not only innovative but also likely to inspire further research in this area. Despite the empirical limitations, the paper's theoretical strengths make it a valuable addition to the field. Therefore, I recommend accepting this paper for a poster presentation.

**Justification For Why Not Higher Score:**

Although the theoretical contributions of the paper are significant, there still remains the empirical limitation of the current paper.

**Justification For Why Not Lower Score:**

The theoretical contributions of the paper are significant. The insights provided about operator preconditioning in the context of training physics-informed models are not only innovative but also likely to inspire further research in this area.

---

### Decision · Program_Chairs · 2024-01-16

Accept (poster)